# ARM: Augment-REINFORCE-Merge Gradient for Stochastic Binary Networks

**Mingzhang Yin**
Department of Statistics and Data Sciences
The University of Texas at Austin
Austin, TX 78712
`mzyin@utexas.edu`

**Mingyuan Zhou**[*]
Department of IROM, McCombs School of Business
The University of Texas at Austin
Austin, TX 78712
`mingyuan.zhou@mccombs.utexas.edu`

## Abstract

To backpropagate the gradients through stochastic binary layers, we propose the augment-REINFORCE-merge (ARM) estimator that is unbiased, exhibits low variance, and has low computational complexity. Exploiting variable augmentation, REINFORCE, and reparameterization, the ARM estimator achieves adaptive variance reduction for Monte Carlo integration by merging two expectations via common random numbers. The variance-reduction mechanism of the ARM estimator can also be attributed to either antithetic sampling in an augmented space, or the use of an optimal anti-symmetric "self-control" baseline function together with the REINFORCE estimator in that augmented space. Experimental results show the ARM estimator provides state-of-the-art performance in auto-encoding variational inference and maximum likelihood estimation, for discrete latent variable models with one or multiple stochastic binary layers. Python code for reproducible research is publicly available.

## 1 Introduction

Given a function $f(\boldsymbol{z})$ of a random variable $\boldsymbol{z} = (z_1, \ldots, z_V)^T$, which follows a distribution $q_{\boldsymbol{\phi}}(\boldsymbol{z})$ parameterized by $\boldsymbol{\phi}$, there has been significant recent interest in estimating $\boldsymbol{\phi}$ to maximize (or minimize) the expectation of $f(\boldsymbol{z})$ with respect to $\boldsymbol{z} \sim q_{\boldsymbol{\phi}}(\boldsymbol{z})$, expressed as

$$\mathcal{E}(\boldsymbol{\phi}) = \int f(\boldsymbol{z}) q_{\boldsymbol{\phi}}(\boldsymbol{z}) d\boldsymbol{z} = \mathbb{E}_{\boldsymbol{z} \sim q_{\boldsymbol{\phi}}(\boldsymbol{z})}[f(\boldsymbol{z})]. \tag{1}$$

In particular, this expectation objective appears in both maximizing the evidence lower bound (ELBO) for variational inference (Jordan et al., 1999; Blei et al., 2017) and approximately maximizing the log marginal likelihood of a hierarchal Bayesian model (Bishop, 1995), two fundamental problems in statistical inference. To maximize (1), if $\nabla_{\boldsymbol{z}} f(\boldsymbol{z})$ is tractable to compute and $\boldsymbol{z} \sim q_{\boldsymbol{\phi}}(\boldsymbol{z})$ can be generated via reparameterization as $\boldsymbol{z} = \mathcal{T}_{\boldsymbol{\phi}}(\boldsymbol{\epsilon})$, $\boldsymbol{\epsilon} \sim p(\boldsymbol{\epsilon})$, where $\boldsymbol{\epsilon}$ are random noises and $\mathcal{T}_{\boldsymbol{\phi}}(\cdot)$ denotes a deterministic transform parameterized by $\boldsymbol{\phi}$, then one may apply the reparameterization trick (Kingma & Welling, 2013; Rezende et al., 2014) to compute the gradient as

$$\nabla_{\boldsymbol{\phi}} \mathcal{E}(\boldsymbol{\phi}) = \nabla_{\boldsymbol{\phi}} \mathbb{E}_{\boldsymbol{\epsilon} \sim p(\boldsymbol{\epsilon})}[f(\mathcal{T}_{\boldsymbol{\phi}}(\boldsymbol{\epsilon}))] = \mathbb{E}_{\boldsymbol{\epsilon} \sim p(\boldsymbol{\epsilon})}[\nabla_{\boldsymbol{\phi}} f(\mathcal{T}_{\boldsymbol{\phi}}(\boldsymbol{\epsilon}))]. \tag{2}$$

This trick, however, is often inapplicable to discrete random variables, as widely used to construct discrete latent variable models such as sigmoid belief networks (Neal, 1992; Saul et al., 1996).

To maximize (1) for discrete $\boldsymbol{z}$, using the score function $\nabla_{\boldsymbol{\phi}} \log q_{\boldsymbol{\phi}}(\boldsymbol{z}) = \nabla_{\boldsymbol{\phi}} q_{\boldsymbol{\phi}}(\boldsymbol{z}) / q_{\boldsymbol{\phi}}(\boldsymbol{z})$, one may compute $\nabla_{\boldsymbol{\phi}} \mathcal{E}(\boldsymbol{\phi})$ via REINFORCE (Williams, 1992) as

$$\nabla_{\boldsymbol{\phi}} \mathcal{E}(\boldsymbol{\phi}) = \mathbb{E}_{\boldsymbol{z} \sim q_{\boldsymbol{\phi}}(\boldsymbol{z})}[f(\boldsymbol{z}) \nabla_{\boldsymbol{\phi}} \log q_{\boldsymbol{\phi}}(\boldsymbol{z})] \approx \frac{1}{K} \sum_{k=1}^{K} f(\boldsymbol{z}^{(k)}) \nabla_{\boldsymbol{\phi}} \log q_{\boldsymbol{\phi}}(\boldsymbol{z}^{(k)}),$$

where $\boldsymbol{z}^{(k)} \overset{iid}{\sim} q_{\boldsymbol{\phi}}(\boldsymbol{z})$ are independent, and identically distributed ($iid$). This unbiased estimator is also known as (a.k.a.) the score-function (Fu, 2006) or likelihood-ratio estimator (Glynn, 1990). While it is unbiased and only requires drawing $iid$ random samples from $q_{\boldsymbol{\phi}}(\boldsymbol{z})$ and computing $\nabla_{\boldsymbol{\phi}} \log q_{\boldsymbol{\phi}}(\boldsymbol{z}^{(k)})$,

---

[*]Corresponding author.

its high Monte-Carlo-integration variance often limits its use in practice. Note that if $f(\boldsymbol{z})$ depends on $\phi$, then we assume it is true that $\mathbb{E}_{\boldsymbol{z} \sim q_\phi(\boldsymbol{z})}[\nabla_\phi f(\boldsymbol{z})] = 0$. For example, in variational inference, we need to maximize the ELBO as $\mathbb{E}_{\boldsymbol{z} \sim q_\phi(\boldsymbol{z})}[f(\boldsymbol{z})]$, where $f(\boldsymbol{z}) = \log[p(\boldsymbol{x} \,|\, \boldsymbol{z})p(\boldsymbol{z})/q_\phi(\boldsymbol{z})]$. In this case, although $f(\boldsymbol{z})$ depends on $\phi$, as $\mathbb{E}_{\boldsymbol{z} \sim q_\phi(\boldsymbol{z})}[\nabla_\phi \log q_\phi(\boldsymbol{z})] = \int \nabla_\phi q_\phi(\boldsymbol{z})d\boldsymbol{z} = \nabla_\phi \int q_\phi(\boldsymbol{z})d\boldsymbol{z} = 0$, we have $\mathbb{E}_{\boldsymbol{z} \sim q_\phi(\boldsymbol{z})}[\nabla_\phi f(\boldsymbol{z})] = 0$.

To address the high-variance issue, one may introduce an appropriate baseline (a.k.a. control variate) to reduce the variance of REINFORCE (Paisley et al., 2012; Ranganath et al., 2014; Mnih & Gregor, 2014; Gu et al., 2016; Mnih & Rezende, 2016; Ruiz et al., 2016; Kucukelbir et al., 2017; Naesseth et al., 2017). Alternatively, one may first relax the discrete random variables with continuous ones and then apply the reparameterization trick to estimate the gradients, which reduces the variance of Monte Carlo integration at the expense of introducing bias (Maddison et al., 2017; Jang et al., 2017). Combining both REINFORCE and the continuous relaxation of discrete random variables, REBAR of Tucker et al. (2017) and RELAX of Grathwohl et al. (2018) both aim to produce a low-variance and unbiased gradient estimator by introducing a continuous relaxation based baseline function, whose parameters, however, need to be estimated at each mini-batch by minimizing the sample variance of the estimator with stochastic gradient descent (SGD). Estimating the baseline parameters often clearly increases the computation. Moreover, the potential conflict, between *minimizing* the sample variance of the gradient estimate and *maximizing* the expectation objective, could slow down or even prevent convergence and increase the risk of overfitting. Another interesting variance-control idea applicable to discrete latent variables is using local expectation gradients, which estimates the gradients based on REINFORCE, by performing Monte Carlo integration using a single global sample together with exact integration of the local variable for each latent dimension (Titsias & Lázaro-Gredilla, 2015).

Distinct from the usual idea of introducing baseline functions and optimizing their parameters to reduce the estimation variance of REINFORCE, we propose the augment-REINFORCE-merge (ARM) estimator, a novel unbiased and low-variance gradient estimator for binary latent variables that is also simple to implement and has low computational complexity. We show by rewriting the expectation with respect to Bernoulli random variables as one with respect to augmented exponential random variables, and then expressing the gradient as an expectation via REINFORCE, one can derive the ARM estimator in the augmented space with the assistance of appropriate reparameterization. In particular, in the augmented space, one can derive the ARM estimator by using either the strategy of sharing common random numbers between two expectations, or the strategy of applying antithetic sampling. Both strategies, as detailedly discussed in Owen (2013), can be used to explain why the ARM estimator is unbiased and could lead to significant variance reduction. Moreover, we show that the ARM estimator can be considered as improving the REINFORCE estimator in an augmented space by introducing an optimal baseline function subject to an anti-symmetric constraint; this baseline function can be considered as a "self-control" one, as it exploits the function $f$ itself and correlated random noises for variance reduction, and adds no extra parameters to learn. This "self-control" feature makes the ARM estimator distinct from both REBAR and RELAX, which rely on minimizing the sample variance of the gradient estimate to optimize the baseline function.

We perform experiments on a representative toy optimization problem and both auto-encoding variational inference and maximum likelihood estimation for discrete latent variable models, with one or multiple binary stochastic layers. Our extensive experiments show that the ARM estimator is unbiased, exhibits low variance, converges fast, has low computation, and provides state-of-the-art out-of-sample prediction performance for discrete latent variable models, suggesting the effectiveness of using the ARM estimator for gradient backpropagation through stochastic binary layers. Python code for reproducible research is available at `https://github.com/mingzhang-yin/ARM-gradient`.

## 2 ARM: AUGMENT-REINFORCE-MERGE ESTIMATOR

In this section, we first present the key theorem of the paper, and then provide its derivation. With this theorem, we summarize ARM gradient ascent for multivariate binary latent variables in Algorithm 1, as shown in Appendix A. Let us denote $\sigma(\phi) = e^\phi/(1 + e^\phi)$ as the sigmoid function and $\mathbf{1}_{[\cdot]}$ as an indicator function that equals to one if the argument is true and zero otherwise.

**Theorem 1** (ARM). *For a vector of $V$ binary random variables $\boldsymbol{z} = (z_1, \ldots, z_V)^T$, the gradient of*

$$\mathcal{E}(\boldsymbol{\phi}) = \mathbb{E}_{\boldsymbol{z} \sim \prod_{v=1}^V \text{Bernoulli}(z_v; \sigma(\phi_v))}[f(\boldsymbol{z})] \tag{3}$$

with respect to $\boldsymbol{\phi} = (\phi_1, \ldots, \phi_V)^T$, *the logits of the Bernoulli probability parameters, can be expressed as*

$$\nabla_{\boldsymbol{\phi}}\mathcal{E}(\boldsymbol{\phi}) = \mathbb{E}_{\boldsymbol{u} \sim \prod_{v=1}^{V} \text{Uniform}(u_v;0,1)} \left[ \left( f(\mathbf{1}_{[\boldsymbol{u}>\sigma(-\boldsymbol{\phi})]}) - f(\mathbf{1}_{[\boldsymbol{u}<\sigma(\boldsymbol{\phi})]}) \right)(\boldsymbol{u} - 1/2) \right], \qquad (4)$$

*where* $\mathbf{1}_{[\boldsymbol{u}>\sigma(-\boldsymbol{\phi})]} := \left( \mathbf{1}_{[u_1>\sigma(-\phi_1)]}, \ldots, \mathbf{1}_{[u_V>\sigma(-\phi_V)]} \right)^T.$

### 2.1 UNIVARIATE ARM ESTIMATOR

Below we will first present the ARM estimator for a univariate binary latent variable (*i.e.*, $V = 1$), and then generalize it to a multivariate one (*i.e.*, $V > 1$). In the univariate case, we need to evaluate the gradient of $\mathcal{E}(\phi) = \mathbb{E}_{z \sim \text{Bernoulli}(\sigma(\phi))}[f(z)]$ with respect to $\phi$, which has an analytic expression as

$$\nabla_\phi \mathcal{E}(\phi) = \nabla_\phi[\sigma(\phi)f(1) + \sigma(-\phi)f(0)] = \sigma(\phi)\sigma(-\phi)[f(1) - f(0)]. \qquad (5)$$

Since $\int_0^{\sigma(\phi)}(1 - 2u)du = \sigma(\phi)\sigma(-\phi)$ and $\int_{\sigma(\phi)}^1(1 - 2u)du = -\sigma(\phi)\sigma(-\phi)$, we can rewrite (5) as

$$\nabla_\phi\mathcal{E}(\phi) = \int_0^{\sigma(\phi)} f(1)(1 - 2u)du + \int_{\sigma(\phi)}^1 f(0)(1 - 2u)du = \int_0^1 f(\mathbf{1}_{[u<\sigma(\phi)]})(1 - 2u)du$$
$$= \mathbb{E}_{u \sim \text{Uniform}(0,1)}[f(\mathbf{1}_{[u<\sigma(\phi)]})(1 - 2u)]. \qquad (6)$$

We refer to (6) as the univariate augment-REINFORCE (AR) estimator, as we initially derived it by combining variable augmentation (Tanner & Wong, 1987; Van Dyk & Meng, 2001), REINFORCE, and Rao Blackwellization (Casella & Robert, 1996); we defer the details to Appendix B.

Applying antithetic sampling (Owen, 2013) to the AR estimator in (6), with $\tilde{u} = 1 - u$, we have

$$\nabla_\phi\mathcal{E}(\phi) = \mathbb{E}_{u \sim \text{Uniform}(0,1)}[f(\mathbf{1}_{[u<\sigma(\phi)]})(1/2 - u)] + \mathbb{E}_{\tilde{u} \sim \text{Uniform}(0,1)}[f(\mathbf{1}_{[\tilde{u}<\sigma(\phi)]})(1/2 - \tilde{u})]$$
$$= \mathbb{E}_{u \sim \text{Uniform}(0,1)}[f(\mathbf{1}_{[u<\sigma(\phi)]})(1/2 - u) + f(\mathbf{1}_{[\tilde{u}<\sigma(\phi)]})(1/2 - \tilde{u})]$$
$$= \mathbb{E}_{u \sim \text{Uniform}(0,1)} \left[ \left( f(\mathbf{1}_{[u>\sigma(-\phi)]}) - f(\mathbf{1}_{[u<\sigma(\phi)]}) \right)(u - 1/2) \right], \qquad (7)$$

which provides the proof for Theorem 1 for $V = 1$. We refer to (7) as the univariate ARM estimator, as we initially derived it by improving the AR estimator with an additional merge step, which shares random numbers between two expectations to reduce the variance of Monte Carlos integration; we defer the details to Appendix B.

Note that since $\mathbb{E}_{u \sim \text{Uniform}(0,1)}[f(\mathbf{1}_{[u<\sigma(\phi)]})(1/2-u)] = -\mathbb{E}_{u \sim \text{Uniform}(0,1)}[f(\mathbf{1}_{[u>\sigma(-\phi)]})(1/2-u)]$, we have $\mathbb{E}_{u \sim \text{Uniform}(0,1)} \left[ \left( f(\mathbf{1}_{[u<\sigma(\phi)]}) + f(\mathbf{1}_{[u>\sigma(-\phi)]}) \right)(1/2 - u) \right] = 0$, subtracted which from the AR estimator in (6) leads to the ARM estimator in (7). For this reason, we can also consider the ARM estimator as the AR estimator subtracted by a zero-mean baseline function as

$$b(u) = \left( f(\mathbf{1}_{[u<\sigma(\phi)]}) + f(\mathbf{1}_{[u>\sigma(-\phi)]}) \right)(1/2 - u).$$

This baseline function is distinct from previously proposed ones in being parameterized by the function $f$ itself over two correlated binary latent variables and satisfying $b(u) = -b(1 - u)$. From this point of view, the ARM estimator can be considered as a "self-control" gradient estimator that exploits the function $f$ itself to control the variance of Monte Carlo integration .

### 2.2 MULTIVARIATE GENERALIZATION VIA THE LAW OF TOTAL EXPECTATION

Although the ARM estimator for univariate binary is of little use in practice, as the true gradient, shown in (5), has an analytic expression, it serves as the foundation for generalizing it to multivariate settings. Let us denote $(\cdot)_{\backslash v}$ as a vector whose $v$th element is removed. For the expectation in (3), applying the univariate ARM estimator in (37) together with the law of total expectation, we have

$$\nabla_{\phi_v}\mathcal{E}(\boldsymbol{\phi}) = \mathbb{E}_{\boldsymbol{z}_{\backslash v} \sim \prod_{j \neq v} \text{Bernoulli}(z_j;\sigma(\phi_j))} \{ \nabla_{\phi_v} \mathbb{E}_{z_v \sim \text{Bernoulli}(\sigma(\phi_v))}[f(\boldsymbol{z})] \}$$
$$= \mathbb{E}_{\boldsymbol{z}_{\backslash v} \sim \prod_{j \neq v} \text{Bernoulli}(z_j;\sigma(\phi_j))} \{ \mathbb{E}_{u_v \sim \text{Uniform}(0,1)} [(u_v - 1/2)$$
$$\times \left( f(\boldsymbol{z}_{\backslash v}, z_v = \mathbf{1}_{[u_v>\sigma(-\phi_v)]}) - f(\boldsymbol{z}_{\backslash v}, z_v = \mathbf{1}_{[u_v<\sigma(\phi_v)]}) \right) ] \}. \qquad (8)$$

Since $\boldsymbol{z}_{\backslash v} \sim \prod_{j \neq v} \text{Bernoulli}(z_j; \sigma(\phi_j))$ can be equivalently generated as $\boldsymbol{z}_{\backslash v} = \mathbf{1}_{[\boldsymbol{u}_{\backslash v}<\sigma(\boldsymbol{\phi}_{\backslash v})]}$ or as $\boldsymbol{z}_{\backslash v} = \mathbf{1}_{[\boldsymbol{u}_{\backslash v}>\sigma(-\boldsymbol{\phi}_{\backslash v})]}$, where $\boldsymbol{u}_{\backslash v} \sim \prod_{j \neq v} \text{Uniform}(u_j; 0, 1)$, exchanging the order of the two

expectations in (8) and applying reparameterization, we conclude the proof for (4) of Theorem 1 with

$$\nabla_{\phi_v}\mathcal{E}(\boldsymbol{\phi}) = \mathbb{E}_{u_v \sim \text{Uniform}(0,1)}\Big\{(u_v - 1/2)\,\mathbb{E}_{\boldsymbol{z}_{\backslash v} \sim \prod_{j \neq v}\text{Bernoulli}(z_j;\sigma(\phi_j))}\Big[$$

$$f(\boldsymbol{z}_{\backslash v}, z_v = \mathbf{1}_{[u_v > \sigma(-\phi_v)]}) - f(\boldsymbol{z}_{\backslash v}, z_v = \mathbf{1}_{[u_v < \sigma(\phi_v)]})\Big]\Big\}$$

$$= \mathbb{E}_{\boldsymbol{u} \sim \prod_{v=1}^V \text{Uniform}(u_v;0,1)}\big[(u_v - 1/2)f(\boldsymbol{z}_{\backslash v} = \mathbf{1}_{[\boldsymbol{u}_{\backslash v} > \sigma(-\boldsymbol{\phi}_{\backslash v})]}, z_v = \mathbf{1}_{[u_v > \sigma(-\phi_v)]})\big]$$

$$- \mathbb{E}_{\boldsymbol{u} \sim \prod_{v=1}^V \text{Uniform}(u_v;0,1)}\big[(u_v - 1/2)f(\boldsymbol{z}_{\backslash v} = \mathbf{1}_{[\boldsymbol{u}_{\backslash v} < \sigma(\boldsymbol{\phi}_{\backslash v})]}, z_v = \mathbf{1}_{[u_v < \sigma(\phi_v)]})\big]$$

$$= \mathbb{E}_{\boldsymbol{u} \sim \prod_{v=1}^V \text{Uniform}(u_v;0,1)}\big[(u_v - 1/2)\big(f(\mathbf{1}_{[\boldsymbol{u} > \sigma(-\boldsymbol{\phi})]}) - f(\mathbf{1}_{[\boldsymbol{u} < \sigma(\boldsymbol{\phi})]})\big)\big]. \tag{9}$$

Alternatively, instead of generalizing the univariate ARM gradient as in (8) and (9), we can first do a multivariate generalization of the univariate AR gradient in (6) as

$$\nabla_{\phi_v}\mathcal{E}(\boldsymbol{\phi}) = \mathbb{E}_{\boldsymbol{z}_{\backslash v} \sim \prod_{j \neq v}\text{Bernoulli}(z_j;\sigma(\phi_j))}\big\{\nabla_{\phi_v}\mathbb{E}_{z_v \sim \text{Bernoulli}(\sigma(\phi_v))}[f(\boldsymbol{z})]\big\}$$

$$= \mathbb{E}_{\boldsymbol{z}_{\backslash v} \sim \prod_{j \neq v}\text{Bernoulli}(z_j;\sigma(\phi_j))}\big\{\mathbb{E}_{u_v \sim \text{Uniform}(0,1)}\big[(1 - 2u_v)f(\boldsymbol{z}_{\backslash v}, z_v = \mathbf{1}_{[u_v < \sigma(\phi_v)]})\big]\big\}$$

$$= \mathbb{E}_{\boldsymbol{u} \sim \prod_{v=1}^V \text{Uniform}(u_v;0,1)}\big[(1 - 2u_v)f(\mathbf{1}_{[\boldsymbol{u} < \sigma(\boldsymbol{\phi})]})\big]. \tag{10}$$

The same as the derivation of the univariate ARM estimator, here we can arrive at (4) from (10) by either adding an antithetic sampling step, or subtracting the AR estimator by a baseline function as

$$\boldsymbol{b}(\boldsymbol{u}) = \big(f(\mathbf{1}_{[\boldsymbol{u} < \sigma(\boldsymbol{\phi})]}) + f(\mathbf{1}_{[\boldsymbol{u} > \sigma(-\boldsymbol{\phi})]})\big)(1/2 - \boldsymbol{u}), \tag{11}$$

which has zero mean, satisfies $\boldsymbol{b}(\boldsymbol{u}) = -\boldsymbol{b}(1 - \boldsymbol{u})$, and is distinct from previously proposed baselines in taking advantage of "self-control" for variance reduction and adding no extra parameters to learn.

## 2.3 EFFECTIVENESS OF ARM FOR VARIANCE REDUCTION

For the univariate case, we show below that the ARM estimator has smaller worst-case variance than REINFORCE does. The proof is deferred to Appendix C.

**Proposition 2** (Univariate gradient variance). *For the objective function* $\mathbb{E}_{z \sim \text{Bernoulli}(\sigma(\phi))}[f(z)]$, *with a single Monte-Carlo sample* $u \sim \text{Uniform}(0,1)$ *or* $z \sim \text{Bernoulli}(\sigma(\phi))$, *the ARM gradient is expressed as* $g_{\text{ARM}}(u, \phi) = \big(f(\mathbf{1}_{[u > \sigma(-\phi)]}) - f(\mathbf{1}_{[u < \sigma(\phi)]})\big)(u - 1/2)$, *and the REINFORCE gradient as* $g_{\text{R}}(z, \phi) = f(z)\nabla_\phi \log \text{Bernoulli}(z;\sigma(\phi)) = f(z)(z - \sigma(\phi))$. *Assuming* $f \geq 0$ *(or* $f \leq 0$), *then* $\frac{\sup_\phi \text{var}[g_{\text{ARM}}(u,\phi)]}{\sup_\phi \text{var}[g_{\text{R}}(u,\phi)]} \leq \frac{16}{25}(1 - 2\frac{f(0)}{f(0) + f(1)})^2 \leq \frac{16}{25}$.

In the general setting, with $\boldsymbol{u}^{(1)}, \ldots, \boldsymbol{u}^{(K)} \overset{iid}{\sim} \prod_{v=1}^V \text{Uniform}(0,1)$, we define the ARM estimate of $\nabla_{\phi_v}\mathcal{E}(\boldsymbol{\phi})$ with $K$ Monte Carlo samples, denoted as $g_{\text{ARM}_K,v}$, and the AR estimate with $2K$ Monte Carlo samples, denoted as $g_{\text{AR}_{2K},v}$, using

$$g_{\text{ARM}_K,v} = \frac{1}{2K}\sum_{k=1}^K (g_v(\boldsymbol{u}^{(k)}) + g_v(1 - \boldsymbol{u}^{(k)})), \quad g_{\text{AR}_{2K},v} = \frac{1}{2K}\sum_{k=1}^{2K} g_v(\boldsymbol{u}^{(k)}), \tag{12}$$

where $g_v(\boldsymbol{u}^{(k)}) = f(\mathbf{1}_{[\boldsymbol{u}^{(k)} < \sigma(\boldsymbol{\phi})]})(1 - 2u_v^{(k)})$. Similar to the analysis in Owen (2013), the amount of variance reduction brought by the ARM estimator can be reflected by the ratio as

$$\frac{\text{var}[g_{\text{ARM}_K,v}]}{\text{var}[g_{\text{AR}_{2K},v}]} = \frac{\text{var}[g_v(\boldsymbol{u})] - \text{Cov}(-g_v(\boldsymbol{u}), g_v(1 - \boldsymbol{u}))}{\text{var}[g_v(\boldsymbol{u})]} = 1 - \rho_v, \ \ \rho_v = \text{Corr}(-g_v(\boldsymbol{u}), g_v(1 - \boldsymbol{u})).$$

Note $-g_v(\boldsymbol{u}) = f(\mathbf{1}_{[\boldsymbol{u} < \sigma(\boldsymbol{\phi})]})(2u_v - 1)$, $g_v(1 - \boldsymbol{u}) = f(\mathbf{1}_{[\boldsymbol{u} > \sigma(-\boldsymbol{\phi})]})(2u_v - 1)$, and $P(\mathbf{1}_{[u_v < \sigma(\phi_v)]} = \mathbf{1}_{[u_v > \sigma(-\phi_v)]}) = \sigma(|\phi_v|) - \sigma(-|\phi_v|)$. Therefore a strong positive correlation (*i.e.*, $\rho_v \to 1$) and hence noticeable variance reduction are likely, especially if $\phi_v$ moves far away from zero during training. Concretely, we have the following proposition.

**Proposition 3** (Variance reduction). *For the ARM estimate* $g_{\text{ARM}_K,v}$ *and AR estimate* $g_{\text{AR}_{2K},v}$ *shown in* (12), *the variance of* $g_{\text{ARM}_K,v}$ *is guaranteed to be lower than that of* $g_{\text{AR}_K,v}$; *moreover, if* $f \geq 0$ *(or* $f \leq 0$), *then the variance of* $g_{\text{ARM}_K,v}$ *is guaranteed to be lower than that of* $g_{\text{AR}_{2K},v}$.

We show below that under the anti-symmetric constraint $\boldsymbol{b}(\boldsymbol{u}) = -\boldsymbol{b}(1 - \boldsymbol{u})$, which implies that $\mathbb{E}_{\boldsymbol{u} \sim \prod_{v=1}^V \text{Uniform}(u_v;0,1)}[\boldsymbol{b}(\boldsymbol{u})]$ is a vector of zeros, Equation (11) is the optimal baseline function to be subtracted from the AR estimator for variance reduction. The proof is deferred to Appendix C.

**Proposition 4** (Optimal anti-symmetric baseline). *For the gradient of $\mathbb{E}_{\boldsymbol{z} \sim q_\phi(\boldsymbol{z})}[f(\boldsymbol{z})]$, the optimal anti-symmetric baseline function to be subtracted from the AR estimator $g_{\mathrm{AR}}(\boldsymbol{u}) = f(\mathbf{1}_{[\boldsymbol{u} < \sigma(\boldsymbol{\phi})]})(1 - 2\boldsymbol{u})$, which minimizes the variance of Monte Carlo integration, can be expressed as*

$$\underset{b_v(\boldsymbol{u}) \in \mathcal{B}}{\arg\min} \, \mathrm{var}[g_{\mathrm{AR},v}(\boldsymbol{u}) - b_v(\boldsymbol{u})] = \frac{1}{2}(g_{\mathrm{AR},v}(\boldsymbol{u}) - g_{\mathrm{AR},v}(1 - \boldsymbol{u})), \tag{13}$$

*where $\mathcal{B} = \{\boldsymbol{b}(\boldsymbol{u}) : b_v(\boldsymbol{u}) = -b_v(1 - \boldsymbol{u}) \text{ for all } v\}$ is the set of all zero-mean anti-symmetric baseline functions. Note the optimal baseline function shown in (13) is exactly the same as (11), which is subtracted from the AR estimator in (10) to arrive at the ARM estimator in (4).*

**Corollary 5** (Lower variance than constant baseline). *The optimal anti-symmetric baseline function for the AR estimator, as shown in (13) and also in (11), leads to lower estimation variance than any constant based baseline function as $b_v(\boldsymbol{u}) = c_v(1/2 - u_v)$, where $c_v$ is a dimension-specific constant whose value can be optimized for variance reduction.*

## 3 BACKPROPAGATION THROUGH DISCRETE STOCHASTIC LAYERS

A latent variable model with multiple stochastic hidden layers can be constructed as

$$\boldsymbol{x} \sim p_{\boldsymbol{\theta}_0}(\boldsymbol{x} \mid \boldsymbol{b}_1), \ \boldsymbol{b}_1 \sim p_{\boldsymbol{\theta}_1}(\boldsymbol{b}_1 \mid \boldsymbol{b}_2), \dots, \boldsymbol{b}_t \sim p_{\boldsymbol{\theta}_t}(\boldsymbol{b}_t \mid \boldsymbol{b}_{t+1}), \dots, \boldsymbol{b}_T \sim p_{\boldsymbol{\theta}_T}(\boldsymbol{b}_T), \tag{14}$$

whose joint likelihood given the distribution parameters $\boldsymbol{\theta}_{0:T} = \{\boldsymbol{\theta}_0, \dots, \boldsymbol{\theta}_T\}$ is expressed as

$$p(\boldsymbol{x}, \boldsymbol{b}_{1:T} \mid \boldsymbol{\theta}_{0:T}) = p_{\boldsymbol{\theta}_0}(\boldsymbol{x} \mid \boldsymbol{b}_1) \Big[ \prod_{t=1}^{T-1} p_{\boldsymbol{\theta}_t}(\boldsymbol{b}_t \mid \boldsymbol{b}_{t+1}) \Big] p_{\boldsymbol{\theta}_T}(\boldsymbol{b}_T). \tag{15}$$

In comparison to deterministic feedforward neural networks, stochastic ones can represent complex distributions and show natural resistance to overfitting (Neal, 1992; Saul et al., 1996; Tang & Salakhutdinov, 2013; Raiko et al., 2014; Gu et al., 2016; Tang & Salakhutdinov, 2013). However, the training of the network, especially if there are stochastic discrete layers, is often much more challenging. Below we show for both auto-encoding variational inference and maximum likelihood estimation, how to apply the ARM estimator for gradient backpropagation in stochastic binary networks.

### 3.1 ARM VARIATIONAL AUTO-ENCODER

For auto-encoding variational inference (Kingma & Welling, 2013; Rezende et al., 2014), we construct a variational distribution as

$$q_{\boldsymbol{w}_{1:T}}(\boldsymbol{b}_{1:T} \mid \boldsymbol{x}) = q_{\boldsymbol{w}_1}(\boldsymbol{b}_1 \mid \boldsymbol{x}) \Big[ \prod_{t=1}^{T-1} q_{\boldsymbol{w}_{t+1}}(\boldsymbol{b}_{t+1} \mid \boldsymbol{b}_t) \Big], \tag{16}$$

with which the ELBO can be expressed as

$$\begin{aligned} \mathcal{E}(\boldsymbol{w}_{1:T}) &= \mathbb{E}_{\boldsymbol{b}_{1:T} \sim q_{\boldsymbol{w}_{1:T}}(\boldsymbol{b}_{1:T} \mid \boldsymbol{x})} \left[ f(\boldsymbol{b}_{1:T}) \right], \ \text{where} \\ f(\boldsymbol{b}_{1:T}) &= \log p_{\boldsymbol{\theta}_0}(\boldsymbol{x} \mid \boldsymbol{b}_1) + \log p_{\boldsymbol{\theta}_{1:T}}(\boldsymbol{b}_{1:T}) - \log q_{\boldsymbol{w}_{1:T}}(\boldsymbol{b}_{1:T} \mid \boldsymbol{x}). \end{aligned} \tag{17}$$

**Proposition 6** (ARM backpropagation). *For a stochastic binary network with $T$ binary stochastic hidden layers, constructing a variational auto-encoder (VAE) defined with $\boldsymbol{b}_0 = \boldsymbol{x}$ and*

$$q_{\boldsymbol{w}_t}(\boldsymbol{b}_t \mid \boldsymbol{b}_{t-1}) = \mathrm{Bernoulli}(\boldsymbol{b}_t; \sigma(\mathcal{T}_{\boldsymbol{w}_t}(\boldsymbol{b}_{t-1}))) \tag{18}$$

*for $t = 1, \dots, T$, the gradient of the ELBO with respect to $\boldsymbol{w}_t$ can be expressed as*

$$\nabla_{\boldsymbol{w}_t} \mathcal{E}(\boldsymbol{w}_{1:T}) = \mathbb{E}_{q(\boldsymbol{b}_{1:t-1})} \left[ \mathbb{E}_{\boldsymbol{u}_t \sim \mathrm{Uniform}(0,1)} [f_\Delta(\boldsymbol{u}_t, \mathcal{T}_{\boldsymbol{w}_t}(\boldsymbol{b}_{t-1}), \boldsymbol{b}_{1:t-1})(\boldsymbol{u}_t - 1/2)] \nabla_{\boldsymbol{w}_t} \mathcal{T}_{\boldsymbol{w}_t}(\boldsymbol{b}_{t-1}) \right],$$

*where* $f_\Delta(\boldsymbol{u}_t, \mathcal{T}_{\boldsymbol{w}_t}(\boldsymbol{b}_{t-1}), \boldsymbol{b}_{1:t-1}) = \mathbb{E}_{\boldsymbol{b}_{t+1:T} \sim q(\boldsymbol{b}_{t+1:T} \mid \boldsymbol{b}_t), \, \boldsymbol{b}_t = \mathbf{1}_{[\boldsymbol{u}_t > \sigma(-\mathcal{T}_{\boldsymbol{w}_t}(\boldsymbol{b}_{t-1}))]}} [f(\boldsymbol{b}_{1:T})]$

$$- \mathbb{E}_{\boldsymbol{b}_{t+1:T} \sim q(\boldsymbol{b}_{t+1:T} \mid \boldsymbol{b}_t), \, \boldsymbol{b}_t = \mathbf{1}_{[\boldsymbol{u}_t < \sigma(\mathcal{T}_{\boldsymbol{w}_t}(\boldsymbol{b}_{t-1}))]}} [f(\boldsymbol{b}_{1:T})]. \tag{19}$$

The gradient presented in (19) can be estimated with a single Monte Carlo sample as

$$\hat{f}_\Delta(\boldsymbol{u}_t, \mathcal{T}_{\boldsymbol{w}_t}(\boldsymbol{b}_{t-1}), \boldsymbol{b}_{1:t-1}) = \begin{cases} 0, & \text{if } \boldsymbol{b}_t^{(1)} = \boldsymbol{b}_t^{(2)}, \\ f(\boldsymbol{b}_{1:t-1}, \boldsymbol{b}_{t:T}^{(1)}) - f(\boldsymbol{b}_{1:t-1}, \boldsymbol{b}_{t:T}^{(2)}), & \text{otherwise}, \end{cases} \tag{20}$$

where $\boldsymbol{b}_t^{(1)} = \mathbf{1}_{[\boldsymbol{u}_t > \sigma(-\mathcal{T}_{\boldsymbol{w}_t}(\boldsymbol{b}_{t-1}))]}$, $\boldsymbol{b}_{t+1:T}^{(1)} \sim q(\boldsymbol{b}_{t+1:T} \mid \boldsymbol{b}_t^{(1)})$, $\boldsymbol{b}_t^{(2)} = \mathbf{1}_{[\boldsymbol{u}_t < \sigma(\mathcal{T}_{\boldsymbol{w}_t}(\boldsymbol{b}_{t-1}))]}$, and $\boldsymbol{b}_{t+1:T}^{(2)} \sim q(\boldsymbol{b}_{t+1:T} \mid \boldsymbol{b}_t^{(2)})$. The proof of Proposition 6 is provided in Appendix C. Suppose the computation complexity of vanilla REINFORCE for a stochastic hidden layer is $\mathcal{O}(1)$, which involves a single evaluation of the function $f$ and gradient backpropagation as $\nabla_{\boldsymbol{w}_t} \mathcal{T}_{\boldsymbol{w}_t}(\boldsymbol{b}_{t-1})$, then for a $T$-stochastic-hidden-layer network, the computation complexity of vanilla REINFORCE is $\mathcal{O}(T)$. By contrast, if evaluating $f$ is much less expensive in computation than gradient backpropagation, then the ARM estimator also has $\mathcal{O}(T)$ complexity, whereas if evaluating $f$ dominates gradient backpropagation in computation, then its worst-case complexity is $\mathcal{O}(2T)$.

## 3.2 ARM MAXIMUM LIKELIHOOD ESTIMATION

For maximum likelihood estimation, the log marginal likelihood can be expressed as

$$\log p_{\boldsymbol{\theta}_{0:T}}(\boldsymbol{x}) = \log \mathbb{E}_{\boldsymbol{b}_{1:T} \sim p_{\boldsymbol{\theta}_{1:T}}(\boldsymbol{b}_{1:T})}[p_{\boldsymbol{\theta}_0}(\boldsymbol{x} \mid \boldsymbol{b}_1)]$$
$$\geq \mathcal{E}(\boldsymbol{\theta}_{1:T}) = \mathbb{E}_{\boldsymbol{b}_{1:T} \sim p_{\boldsymbol{\theta}_{1:T}}(\boldsymbol{b}_{1:T})}[\log p_{\boldsymbol{\theta}_0}(\boldsymbol{x} \mid \boldsymbol{b}_1)]. \tag{21}$$

Generalizing Proposition 6 leads to the following proposition.

**Proposition 7.** *For a stochastic binary network defined as*

$$p_{\boldsymbol{\theta}_t}(\boldsymbol{b}_t \mid \boldsymbol{b}_{t+1}) = \text{Bernoulli}(\boldsymbol{b}_t; \sigma(\mathcal{T}_{\boldsymbol{\theta}_t}(\boldsymbol{b}_{t+1}))), \tag{22}$$

*the gradient of the lower bound in* (21) *with respect to $\boldsymbol{\theta}_t$ can be expressed as*

$$\nabla_{\boldsymbol{\theta}_t} \mathcal{E}(\boldsymbol{\theta}_{1:T}) = \mathbb{E}_{p(\boldsymbol{b}_{t+1:T})} \left[ \mathbb{E}_{\boldsymbol{u}_t \sim \text{Uniform}(0,1)} [f_\Delta(\boldsymbol{u}_t, \mathcal{T}_{\boldsymbol{\theta}_t}(\boldsymbol{b}_{t+1}), \boldsymbol{b}_{t+1:T})(\boldsymbol{u}_t - 1/2)] \nabla_{\boldsymbol{\theta}_t} \mathcal{T}_{\boldsymbol{\theta}_t}(\boldsymbol{b}_{t+1}) \right],$$

*where* $f_\Delta(\boldsymbol{u}_t, \mathcal{T}_{\boldsymbol{\theta}_t}(\boldsymbol{b}_{t+1}), \boldsymbol{b}_{t+1:T}) = \mathbb{E}_{\boldsymbol{b}_{1:t-1} \sim p(\boldsymbol{b}_{1:t-1} \mid \boldsymbol{b}_t), \, \boldsymbol{b}_t = \mathbf{1}_{[\boldsymbol{u}_t > \sigma(-\mathcal{T}_{\boldsymbol{\theta}_t}(\boldsymbol{b}_{t+1}))]}}[\log p_{\boldsymbol{\theta}_0}(\boldsymbol{x} \mid \boldsymbol{b}_1)]$

$$- \mathbb{E}_{\boldsymbol{b}_{1:t-1} \sim p(\boldsymbol{b}_{1:t-1} \mid \boldsymbol{b}_t), \, \boldsymbol{b}_t = \mathbf{1}_{[\boldsymbol{u}_t < \sigma(\mathcal{T}_{\boldsymbol{\theta}_t}(\boldsymbol{b}_{t+1}))]}}[\log p_{\boldsymbol{\theta}_0}(\boldsymbol{x} \mid \boldsymbol{b}_1)].$$

## 4 EXPERIMENTAL RESULTS

To illustrate the working mechanism of the ARM estimator, related to Tucker et al. (2017) and Grathwohl et al. (2018), we consider learning $\phi$ to maximize

$$\mathcal{E}(\phi) = \mathbb{E}_{z \sim \text{Bernoulli}(\sigma(\phi))}[(z - p_0)^2], \text{ where } p_0 \in \{0.49, 0.499, 0.501, 0.51\}.$$

The optimal solution is $\sigma(\phi) = \mathbf{1}_{[p_0 < 0.5]}$. The closer $p_0$ is to 0.5, the more challenging the optimization becomes. We compare both the AR and ARM estimators to the true gradient as

$$g_\phi = (1 - 2p_0)\sigma(\phi)(1 - \sigma(\phi)) \tag{23}$$

and three previously proposed unbiased estimators, including REINFORCE, REBAR (Tucker et al., 2017), and RELAX (Grathwohl et al., 2018). Since RELAX is closely related to REBAR in introducing stochastically estimated control variates to improve REINFORCE, and clearly outperforms RE-BAR in our experiments for this toy problem (as also shown in Grathwohl et al. (2018) for $p_0 = 0.49$), we omit the results of REBAR for brevity. With a single random sample $u \sim \text{Uniform}(0,1)$ for Monte Carlo integration, the REINFORCE and AR gradients can be expressed as

$$g_{\phi,\text{REINFORCE}} = (\mathbf{1}_{[u < \sigma(\phi)]} - p_0)^2(\mathbf{1}_{[u < \sigma(\phi)]} - \sigma(\phi)), \qquad g_{\phi,\text{AR}} = (\mathbf{1}_{[u < \sigma(\phi)]} - p_0)^2(1 - 2u),$$

while the ARM gradient can be expressed as

$$g_{\phi,\text{ARM}} = \left[ (\mathbf{1}_{[u > \sigma(-\phi)]} - p_0)^2 - (\mathbf{1}_{[u < \sigma(\phi)]} - p_0)^2 \right] (u - 1/2). \tag{24}$$

See Grathwohl et al. (2018) for the details on RELAX.

As shown in Figure 1, the REINFORCE gradients have large variances. Consequently, a REINFORCE based gradient ascent algorithm may diverge if the gradient ascent stepsize is not sufficiently small. For example, when $p_0 = 0.501$, the optimal value for the Bernoulli probability $\sigma(\phi)$ is 0, but the algorithm with 0.1 as the stepsize infers it to be close to 1 at the end of 2000 iterations of a random trial. The AR estimator behaves similarly as REINFORCE does. By contrast, both RELAX and ARM exhibit clearly lower estimation variance. It is interesting to note that the trace plots of the estimated probability $\sigma(\phi)$ with the univariate ARM estimator almost exactly match these with the

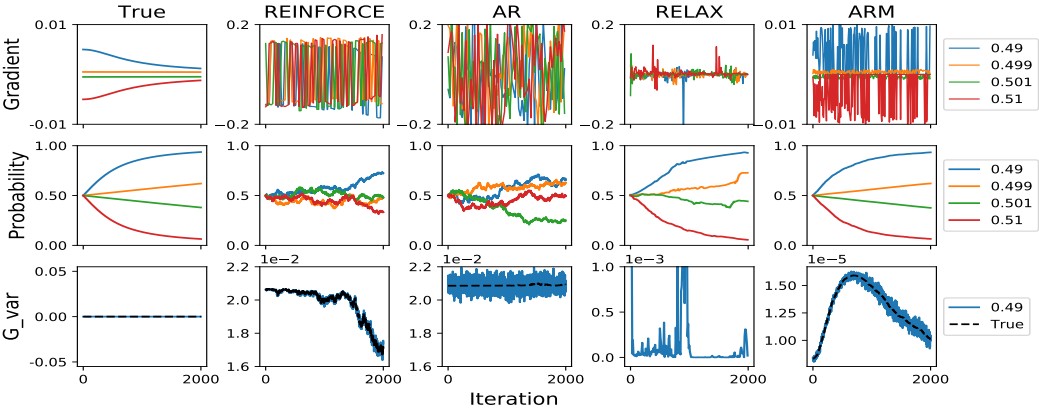

Figure 1: Comparison of a variety of gradient estimators in maximizing $\mathcal{E}(\phi) = \mathbb{E}_{z \sim \text{Bernoulli}(\sigma(\phi))}[(z - p_0)^2]$ via gradient ascent, where $p_0 \in \{0.49, 0.499, 0.501, 0.51\}$; the optimal solution is $\sigma(\phi) = \mathbf{1}(p_0 < 0.5)$. Shown in Rows 1 and 2 are the trace plots of the true/estimated gradients $\nabla_\phi \mathcal{E}(\phi)$ and estimated Bernoulli probability parameters $\sigma(\phi)$, with $\phi$ updated via gradient ascent. Shown in Row 3 are the gradient variances for $p_0 = 0.49$, estimated using $K = 5000$ Monte Carlo samples at each iteration; the theoretical gradient variances are also shown if they can be analytically calculated (see Appendices C and D and for related analytic expressions).

true gradients, despite that the trace plots of the ARM gradients are distinct from these of the true gradients. More specifically, while the true gradients smoothly evolve over iterations, the univariate ARM gradients are characterized by zeros and random spikes; this distinct behavior is expected by examining (38) in Appendix C, which suggests that at any given iteration, the univariate ARM gradient based on a single Monte Carlo sample is either exactly zero, which happens with probability $\sigma(|\phi|) - \sigma(-|\phi|)$, or taking $|[f(1) - f(0)](1/2 - u)|$ as its absolute value. These observations suggest that by adjusting the frequencies and amplitudes of spike gradients, the univariate ARM estimator very well approximates the behavior of the true gradient for learning with gradient ascent.

In Figure 4 of Appendix D, we plot the gradient estimated with multiple Monte Carlo samples against the true gradient at each iteration, further showing the ARM estimator has the lowest estimation variance given the same number of Monte Carlo samples. Moreover, in Figure 5 of Appendix D, for each estimator specific column, we plot against the value of $\phi$ the sample mean $\bar{g}$, sample standard deviation $s_g$, and the gradient signal-to-noise ratio defined as $\text{SNR}_g = |\bar{g}|/s_g$; for each $\phi$ value, we use $K = 1000$ single-Monte-Carlo-sample gradient estimates to calculate $\bar{g}$, $s_g$, and $\text{SNR}_g$. Both figures further show that the ARM estimator outperforms not only REINFORCE, which has large variance, but also RELAX, which improves REINFORCE with an adaptively estimated baseline.

In Figure 5 of Appendix D, it is also interesting to notice that the gradient signal-to-noise ratio for the ARM estimator appears to be only a function of $\phi$ but not a function of $p_0$; this can be verified to be true using (23) and (39) in Appendix C, as the ratio of the absolute value of the true gradient $|g_\phi|$ to $\sqrt{\text{var}[g_{\phi,\text{ARM}}]}$, the standard deviation of the ARM estimate in (24), can be expressed as

$$\sigma(\phi)(1 - \sigma(\phi)) / \sqrt{\tfrac{1}{16}(1 - t)(t^3 + \tfrac{7}{3}t^2 + \tfrac{1}{3}t + \tfrac{1}{3})}, \ \ t = \sigma(|\phi|) - \sigma(-|\phi|). \tag{25}$$

We find that the values of the ratio shown above are almost exactly matched by the values of $\text{SNR}_g = |\bar{g}|/s_g$ under the ARM estimator, shown in the bottom right subplot of Figure 5. Therefore, for this example optimization problem, the ARM estimator exhibits a desirable property in providing high gradient signal-to-noise ratios regardless of the value of $p_0$.

## 4.1 DISCRETE VARIATIONAL AUTO-ENCODERS

To optimize a variational auto-encoder (VAE) for a discrete latent variable model, existing solutions often rely on biased but low-variance stochastic gradient estimators (Bengio et al., 2013; Jang et al., 2017), unbiased but high-variance ones (Mnih & Gregor, 2014), or unbiased REINFORCE combined with computationally expensive baselines, whose parameters are estimated by minimizing the sample variance of the estimator with SGD (Tucker et al., 2017; Grathwohl et al., 2018). By contrast, the ARM estimator exhibits low variance and is unbiased, efficient to compute, and simple to implement.

Table 1: The constructions of three differently structured discrete variational auto-encoders. The following symbols "→", "]", )", and "↝" represent deterministic linear transform, leaky rectified linear units (LeakyReLU) (Maas et al., 2013) nonlinear activation, sigmoid nonlinear activation, and random sampling respectively, in the encoder (a.k.a. recognition network); their reversed versions are used in the decoder (a.k.a. generator).

|         | Nonlinear | Linear | Linear two layers |
|---------|-----------|--------|-------------------|
| Encoder | 784→200]→200]→200)↝200 | 784→200)↝200 | 784→200)↝200→200)↝200 |
| Decoder | 784↜(784←[200←[200←200 | 784↜(784←200 | 784↜(784←200↜(200←200 |

Table 2: Test negative log-likelihoods of discrete VAEs trained with a variety of stochastic gradient estimators on MNIST-static and OMNIGLOT, where ∗, ⋆, †, ‡ represent the results reported in Mnih & Gregor (2014), Tucker et al. (2017), Gu et al. (2016), and Grathwohl et al. (2018), respectively. The results for LeGrad (Titsias & Lázaro-Gredilla, 2015) are obtained by running the code provided by the authors. We report the results of ARM using the sample mean and standard deviation over five independent trials with random initializations.

(a) MNIST-static

| Linear | | Nonlinear | | Two layers | |
|--------|--------|-----------|--------|-----------|--------|
| Algorithm | $-\log p(x)$ | Algorithm | $-\log p(x)$ | Algorithm | $-\log p(x)$ |
| AR | $= 164.1$ | AR | $= 114.6$ | AR | $= 162.2$ |
| REINFORCE | $= 170.1$ | REINFORCE | $= 114.1$ | REINFORCE | $= 159.2$ |
| Wake-Sleep[∗] | $= 120.8$ | Wake-Sleep[∗] | - | Wake-Sleep[∗] | $= 107.7$ |
| NVIL [∗] | $= 113.1$ | NVIL [∗] | $= 102.2$ | NVIL[∗] | $= 99.8$ |
| LeGrad | $\leq 117.5$ | LeGrad | - | LeGrad | - |
| MuProp[†] | $\leq 113.0$ | MuProp[⋆] | $= 99.1$ | MuProp[†] | $\leq 100.4$ |
| Concrete[⋆] | $= 107.2$ | Concrete[⋆] | $= 99.6$ | Concrete[⋆] | $= 95.6$ |
| REBAR[⋆] | $= 107.7$ | REBAR[⋆] | $= 100.7$ | REBAR[⋆] | $= \mathbf{95.7}$ |
| RELAX[‡] | $\leq 113.6$ | RELAX[‡] | $\leq 119.2$ | RELAX[‡] | $\leq 100.9$ |
| ARM | $= \mathbf{107.2 \pm 0.1}$ | ARM | $= \mathbf{98.4 \pm 0.3}$ | ARM | $= 96.7 \pm 0.3$ |

(b) OMNIGLOT

| Linear | | Nonlinear | | Two layers | |
|--------|--------|-----------|--------|-----------|--------|
| Algorithm | $-\log p(x)$ | Algorithm | $-\log p(x)$ | Algorithm | $-\log p(x)$ |
| NVIL[∗] | $= 117.6$ | NVIL[∗] | $= \mathbf{116.6}$ | NVIL[∗] | $= 111.4$ |
| MuProp[⋆] | $= 117.6$ | MuProp[⋆] | $= 117.5$ | MuProp[⋆] | $= 111.2$ |
| Concrete[⋆] | $= 117.7$ | Concrete[⋆] | $= 116.7$ | Concrete[⋆] | $= 111.3$ |
| REBAR[⋆] | $= 117.7$ | REBAR[⋆] | $= 118.0$ | REBAR[⋆] | $= 110.8$ |
| RELAX[‡] | $\leq 122.1$ | RELAX[‡] | $\leq 128.2$ | RELAX[‡] | $\leq 115.4$ |
| ARM | $= \mathbf{115.8 \pm 0.2}$ | ARM | $= 117.6 \pm 0.4$ | ARM | $= \mathbf{109.8 \pm 0.3}$ |

For discrete VAEs, we compare ARM with a variety of representative stochastic gradient estimators for discrete latent variables, including Wake-Sleep (Hinton et al., 1995), NVIL (Mnih & Gregor, 2014), LeGrad (Titsias & Lázaro-Gredilla, 2015), MuProp (Gu et al., 2016), Concrete (Gumbel-Softmax) (Jang et al., 2017; Maddison et al., 2017), REBAR (Grathwohl et al., 2018), and RELAX (Tucker et al., 2017). Following the settings in Tucker et al. (2017) and Grathwohl et al. (2018), for the encoder defined in (15) and decoder defined in (16), we consider three different network architectures, as summarized in Table 1, including "Nonlinear" that has one stochastic but two Leaky-ReLU (Maas et al., 2013) deterministic hidden layers, "Linear" that has one stochastic hidden layer, and "Linear two layers" that has two stochastic hidden layers. We consider a widely used binarization (Salakhutdinov & Murray, 2008; Larochelle & Murray, 2011; Yin & Zhou, 2018), referred to as MNIST-static and available at http://www.dmi.usherb.ca/~larocheh/mlpython/_modules/datasets/binarized_mnist.html, making our numerical results directly comparable to those reported in the literature. In addition to MNIST-static, we also consider MNIST-threshold (van den Oord et al., 2017), which binarizes MNIST by thresholding each pixel value at 0.5, and the binarized OMNIGLOT dataset.

We train discrete VAEs with 200 conditionally $iid$ Bernoulli random variables as the hidden units of each stochastic binary layer. We maximize a single-Monte-Carlo-sample ELBO using Adam (Kingma & Ba, 2014), with the learning rate selected from $\{5, 1, 0.5\} \times 10^{-4}$ by the validation set. We set the batch size as 50 for MNIST and 25 for OMNIGLOT. For each dataset, using its default

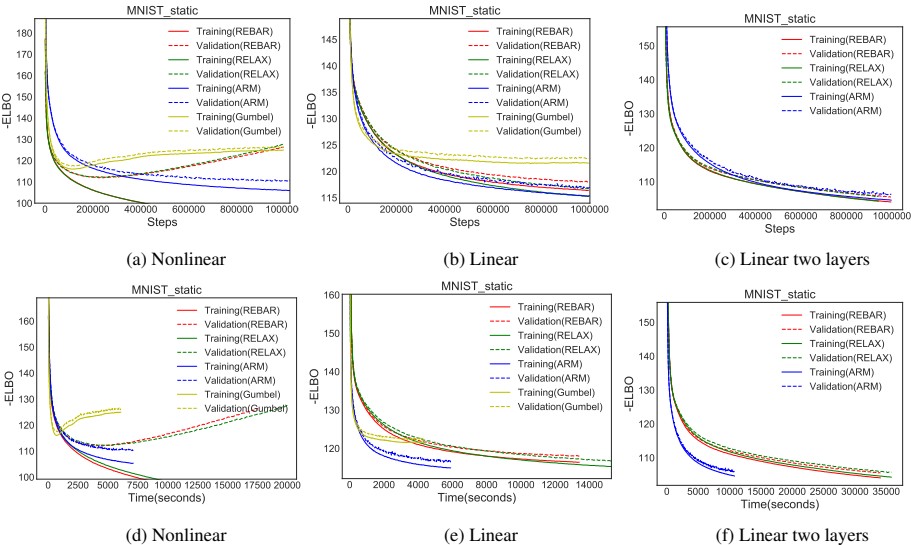

(a) Nonlinear      (b) Linear      (c) Linear two layers

(d) Nonlinear      (e) Linear      (f) Linear two layers

Figure 2: Training and validation negative ELBOs on MNIST-static with respect to the training iterations, shown in the top row, and with respect to the wall clock times on Tesla-K40 GPU, shown in the bottom row, for three differently structured Bernoulli VAEs.

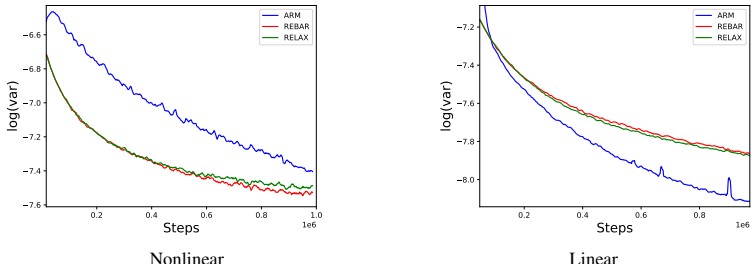

Nonlinear          Linear

Figure 3: Trace plots of the log variance of the gradient estimators on the MNIST-static data for "Nonlinear" and "Linear" network architectures, whose corresponding trace plots of both the training and validation -ELBO are shown in Figures 2(a) and 2(b), respectively. The variance of the gradient of each element is estimated by performing exponential smoothing, with the smoothing factor as 0.999, on its first two moments; the logarithm of the average over all elements' gradient variances is shown for each stochastic gradient ascent step.

training/validation/testing partition, we train all methods on the training set, calculate the validation log-likelihood for every epoch, and report the test negative log-likelihood when the validation negative log-likelihood reaches its minimum within a predefined maximum number of iterations.

We summarize the test negative log-likelihoods in Table 2 for MNIST-static. We also summarize the test negative ELBOs in Table 4 of the Appendix, and provide related trace plots of the training and validation negative ELBOs on MNIST-static in Figure 2, and these on MNIST-threshold and OMNIGLOT in Figures 6 and 7 of the Appendix, respectively. For these trace plots, for a fair comparison of convergence speed between different algorithms, we use publicly available code and setting the learning rate of ARM the same as that selected by RELAX in Grathwohl et al. (2018). Note as shown in Figures 2(a,d) and 7(a,d), both REBAR and RELAX exhibit clear signs of overfitting on both MNIST-static and Omniglot using the "Nonlinear" architecture; as ARM runs much faster per iteration than both of them and do not exhibit overfitting given the same number of iterations, we allow ARM to run more stochastic gradient ascent steps under these two scenarios to check whether it will eventually overfit the training set.

These results show that ARM provides state-of-the-art performance in delivering not only fast convergence, but also low negative log-likelihoods and negative ELBOs on both the validation and test sets, with low computational cost, for all three different network architectures. In comparison to the vanilla REINFORCE on MNIST-static, as shown in Table 2 (a), ARM achieves significantly lower test negative log-likelihoods, which can be explained by having much lower variance in its gradient

Table 3: Comparison of the test negative log-likelihoods between ARM and various gradient estimators in Jang et al. (2017), for the MNIST conditional distribution estimation benchmark task.

| Gradient estimator | ARM | ST | DARN | Annealed ST | ST Gumbel-S. | SF | MuProp |
|---|---|---|---|---|---|---|---|
| $-\log p(\boldsymbol{x}_l \,|\, \boldsymbol{x}_u)$ | $\mathbf{57.9 \pm 0.1}$ | 58.9 | 59.7 | 58.7 | 59.3 | 72.0 | 58.9 |

estimation, while only costing 20% to 30% more computation time to finish the same number of iterations.

The trace plots in Figures 2, 6, and 7 show that ARM achieves its objective better or on a par with the competing methods in all three different network architectures. In particular, the performance of ARM on MNIST-threshold is significantly better, suggesting ARM is more robust, better resists overfitting, and has better generalization ability. On both MNIST-static and OMNIGLOT, with the "Nonlinear" network architecture, both REBAR and RELAX exhibit severe overfitting, which could be caused by their training procedure, which updates the parameters of the baseline function by minimizing the sample variance of the gradient estimator using SGD. For less overfitting linear and two-stochastic-layer networks, ARM overall performs better than both REBAR and RELAX and runs significantly faster (about 6-8 times faster) in terms of the computation time per iteration.

To understand why ARM has the best overall performance, we examine the trace plots of the logarithm of the estimated variance of gradient estimates in Figure 3. On the MNIST-static dataset with the "Nonlinear" network, the left subplot of Figure 3 shows that both REBAR and RELAX exhibit lower variance than ARM does for their single-Monte-Carlo-sample based gradient estimates; however, the corresponding trace plots of the validation negative ELBOs, shown in Figure 2(a), suggest they both severely overfit the training data as the learning progresses; our hypothesis for this phenomenon is that REBAR and RELAX may favor suboptimal solutions that are associated with lower gradient variance; in other words, they may have difficulty in converging to local optimal solutions that are associated with high gradient variance. For the "Linear" network architecture, the right subplot of Figure 3 shows that ARM exhibits lower variance for its gradient estimate than both REBAR and RELAX do, and Figure 2(b) shows that none of them exhibit clear signs of overfitting; this observation could be used to explain why ARM results in the best convergence for both the training and validation negative ELBOs, as shown in Figure 2(b).

### 4.2 Maximum likelihood estimation for a stochastic binary network

Denoting $\boldsymbol{x}_l, \boldsymbol{x}_u \in \mathbb{R}^{394}$ as the lower and upper halves of an MNIST digit, respectively, we consider a standard benchmark task of estimating the conditional distribution $p_{\boldsymbol{\theta}_{0:2}}(\boldsymbol{x}_l \,|\, \boldsymbol{x}_u)$ (Raiko et al., 2014; Bengio et al., 2013; Gu et al., 2016; Jang et al., 2017; Tucker et al., 2017), using a stochastic binary network with two stochastic binary hidden layers, expressed as

$$\boldsymbol{x}_l \sim \text{Bernoulli}(\sigma(\mathcal{T}_{\boldsymbol{\theta}_0}(\boldsymbol{b}_1))), \ \boldsymbol{b}_1 \sim \text{Bernoulli}(\sigma(\mathcal{T}_{\boldsymbol{\theta}_1}(\boldsymbol{b}_2))), \ \boldsymbol{b}_2 \sim \text{Bernoulli}(\sigma(\mathcal{T}_{\boldsymbol{\theta}_2}(\boldsymbol{x}_u))). \quad (26)$$

We set the network structure as 392-200-200-392, which means both $\boldsymbol{b}_1$ and $\boldsymbol{b}_2$ are 200 dimensional binary vectors and the transformation $\mathcal{T}_{\boldsymbol{\theta}}$ are linear so the results are directly comparable with those in Jang et al. (2017). We approximate $\log p_{\boldsymbol{\theta}_{0:2}}(\boldsymbol{x}_l \,|\, \boldsymbol{x}_u)$ with $\log \frac{1}{K} \sum_{k=1}^{K} \text{Bernoulli}(\boldsymbol{x}_l; \sigma(\mathcal{T}_{\boldsymbol{\theta}_0}(\boldsymbol{b}_1^{(k)})))$, where $\boldsymbol{b}_1^{(k)} \sim \text{Bernoulli}(\sigma(\mathcal{T}_{\boldsymbol{\theta}_1}(\boldsymbol{b}_2^{(k)})))$, $\boldsymbol{b}_2^{(k)} \sim \text{Bernoulli}(\sigma(\mathcal{T}_{\boldsymbol{\theta}_2}(\boldsymbol{x}_u)))$. We perform training with $K = 1$, which can also be considered as optimizing on a single-Monte-Carlo-sample estimate of the lower bound of the log marginal likelihood shown in (21). We use Adam (Kingma & Ba, 2014), with the learning rate set as $10^{-4}$, mini-batch size as 100, and number of epochs for training as 2000. Given the inferred point estimate of $\boldsymbol{\theta}_{0:2}$ after training, we evaluate the accuracy of conditional density estimation by estimating the negative log-likelihood as $-\log p_{\boldsymbol{\theta}_{0:2}}(\boldsymbol{x}_l \,|\, \boldsymbol{x}_u)$, averaging over the test set using $K = 1000$. We show example results of predicting the activation probabilities of the pixels of $\boldsymbol{x}_l$ given $\boldsymbol{x}_u$ in Figure 8 of the Appendix.

As shown in Table 3, optimizing a stochastic binary network with the ARM estimator, which is unbiased and computationally efficient, achieves the lowest test negative log-likelihood, outperforming previously proposed biased stochastic gradient estimators on similarly structured stochastic networks, including DARN (Gregor et al., 2013), straight through (ST) (Bengio et al., 2013), slope-annealed ST (Chung et al., 2016), and ST Gumbel-softmax (Jang et al., 2017), and unbiased ones, including score-function (SF) and MuProp (Gu et al., 2016).

## 5 Conclusion

To train a discrete latent variable model with one or multiple stochastic binary layers, we propose the augment-REINFORCE-merge (ARM) estimator to provide unbiased and low-variance gradient estimates of the parameters of Bernoulli distributions. With a single Monte Carlo sample, the estimated gradient is the product of uniform random noises and the difference of a function of two vectors of correlated binary latent variables. Without relying on estimating a baseline function with extra learnable parameters for variance reduction, it maintains efficient computation and avoids increasing the risk of overfitting. Applying the ARM gradient leads to not only fast convergence, but also low test negative log-likelihoods (and low test negative evidence lower bounds for variational inference), on both auto-encoding variational inference and maximum likelihood estimation for stochastic binary feedforward neural networks. Some natural extensions of the proposed ARM estimator include generalizing it to multivariate categorical latent variables, combining it with a baseline or local-expectation based variance reduction method, and applying it to reinforcement learning whose action space is discrete.

## Acknowledgments

M. Zhou acknowledges the support of Award IIS-1812699 from the U.S. National Science Foundation, and support from the McCombs Research Excellence Grant. The authors acknowledge the support of NVIDIA Corporation with the donation of the Titan Xp GPU used for this research, and the computational support of Texas Advanced Computing Center.

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

# Appendix

## A  THE ARM GRADIENT ASCENT ALGORITHM

We summarize the algorithm to compute ARM gradient for binary latent variables. Here we show the gradient with respect to the logits associated with the probability of Bernoulli random variables. If the logits are further generated by deterministic transform such as neural networks, the gradient with respect to the transform parameters can be directly computed by the chain rule. For stochastic transforms, the implementation of ARM gradient is discussed in detail in Section 3 and summarized in Algorithm 2.

---

**Algorithm 1:** ARM gradient for a $V$-dimensional binary latent vector

**input**  : Bernoulli distribution $\{q_{\phi_v}(z_v)\}_{v=1:V}$ with probability $\{\sigma(\phi_v)\}_{v=1:V}$, target $f(\boldsymbol{z})$;
$\quad\quad \boldsymbol{z} = (z_1, \cdots, z_V), \boldsymbol{\phi} = (\phi_1, \cdots, \phi_V)$
**output**: $\boldsymbol{\phi}$ and $\boldsymbol{\psi}$ that maximize $\mathcal{E}(\boldsymbol{\phi}, \boldsymbol{\psi}) = \mathbb{E}_{\boldsymbol{z} \sim \prod_{v=1}^{V} q_{\phi_v}(z_v)}[f(\boldsymbol{z}; \boldsymbol{\psi})]$

Initialize $\boldsymbol{\phi}, \boldsymbol{\psi}$ randomly;
**while** *not converged* **do**
$\quad$ Sample a mini-batch of $\boldsymbol{x}$ from the data;
$\quad$ Sample $z_v \sim \text{Bernoulli}(\sigma(\phi_v))$ for $v = 1, \cdots, V$ ;
$\quad$ sample $u_v \sim \text{Uniform}(0, 1)$ for $v = 1, \cdots, V, \boldsymbol{u} = (u_1, \cdots, u_V)$ ;
$\quad$ $g_{\boldsymbol{\psi}} = \nabla_{\boldsymbol{\psi}} f(\boldsymbol{z}; \boldsymbol{\psi})$ ;
$\quad$ $f_\Delta(\boldsymbol{u}, \boldsymbol{\phi}) = f\big(\mathbf{1}_{[\boldsymbol{u} > \sigma(-\boldsymbol{\phi})]}\big) - f\big(\mathbf{1}_{[\boldsymbol{u} < \sigma(\boldsymbol{\phi})]}\big)$ ;
$\quad$ $g_{\boldsymbol{\phi}} = f_\Delta(\boldsymbol{u}, \boldsymbol{\phi})(\boldsymbol{u} - 0.5)$
$\quad$ $\boldsymbol{\phi} = \boldsymbol{\phi} + \rho_t g_{\boldsymbol{\phi}}, \quad \boldsymbol{\psi} = \boldsymbol{\psi} + \eta_t g_{\boldsymbol{\psi}}, \quad$ with stepsizes $\rho_t, \eta_t$
**end**

---

**Algorithm 2:** ARM gradient for a $T$-stochastic-hidden-layer binary network

**input**  : Inference network $q_{\boldsymbol{w}_{1:T}}(\boldsymbol{b}_{1:T} \mid \boldsymbol{x}) = q_{\boldsymbol{w}_1}(\boldsymbol{b}_1 \mid \boldsymbol{x})\Big[\prod_{t=1}^{T-1} q_{\boldsymbol{w}_{t+1}}(\boldsymbol{b}_{t+1} \mid \boldsymbol{b}_t)\Big]$ where
$\quad\quad q_{\boldsymbol{w}_t}(\boldsymbol{b}_t \mid \boldsymbol{b}_{t-1}) = \text{Bernoulli}(\boldsymbol{b}_t; \sigma(\mathcal{T}_{\boldsymbol{w}_t}(\boldsymbol{b}_{t-1})))$, target $f(\boldsymbol{b}_{1:T}; \boldsymbol{\psi})$
**output**: $\boldsymbol{w}_{1:T}$ and $\boldsymbol{\psi}$ that maximize $\mathcal{E}(\boldsymbol{w}_{1:T}, \boldsymbol{\psi}) = \mathbb{E}_{\boldsymbol{b}_{1:T} \sim q_{\boldsymbol{w}_{1:T}}}[f(\boldsymbol{b}_{1:T}; \boldsymbol{\psi})]$

Initialize $\boldsymbol{w}_{1:T}, \boldsymbol{\psi}$ randomly;
**while** *not converged* **do**
$\quad$ Sample a mini-batch of $\boldsymbol{x}$ from data;
$\quad$ **for** *t = 1:T* **do**
$\quad\quad$ If $t \geq 2$, sample $\boldsymbol{b}_{t-1} \sim q(\boldsymbol{b}_{t-1}|\boldsymbol{b}_{t-2})$, if $t = 2, \boldsymbol{b}_{1:t-1} = \boldsymbol{b}_1$, else $\boldsymbol{b}_{1:t-1} = [\boldsymbol{b}_{1:t-2}, \boldsymbol{b}_{t-1}]$ ;
$\quad\quad$ sample $\boldsymbol{u}_t \sim \prod \text{Uniform}(0, 1)$ ;
$\quad\quad$ $\boldsymbol{b}_t^1 = \mathbf{1}_{[\boldsymbol{u}_t > \sigma(-\mathcal{T}_{\boldsymbol{w}_t}(\boldsymbol{b}_{t-1}))]}$ ;
$\quad\quad$ $\boldsymbol{b}_t^2 = \mathbf{1}_{[\boldsymbol{u}_t < \sigma(\mathcal{T}_{\boldsymbol{w}_t}(\boldsymbol{b}_{t-1}))]}$ ;
$\quad\quad$ **if** $\boldsymbol{b}_t^1 = \boldsymbol{b}_t^2$ **then**
$\quad\quad\quad$ $f_\Delta(\boldsymbol{b}_{1:t-1}, \boldsymbol{b}_{t:T}^1, \boldsymbol{b}_{t:T}^2) = 0$ ;
$\quad\quad$ **else**
$\quad\quad\quad$ $\boldsymbol{b}_{t+1:T}^1 \sim q(\boldsymbol{b}_{t+1:T}|\boldsymbol{b}_t^1)$ ;
$\quad\quad\quad$ $\boldsymbol{b}_{t+1:T}^2 \sim q(\boldsymbol{b}_{t+1:T}|\boldsymbol{b}_t^2)$ ;
$\quad\quad\quad$ $f_\Delta(\boldsymbol{b}_{1:t-1}, \boldsymbol{b}_{t:T}^1, \boldsymbol{b}_{t:T}^2) = f(\boldsymbol{b}_{1:t-1}, \boldsymbol{b}_{t:T}^1) - f(\boldsymbol{b}_{1:t-1}, \boldsymbol{b}_{t:T}^2)$ ;
$\quad\quad\quad$ $g_{\boldsymbol{w}_t} = f_\Delta(\boldsymbol{b}_{1:t-1}, \boldsymbol{b}_{t:T}^1, \boldsymbol{b}_{t:T}^2)(\boldsymbol{u}_t - \frac{1}{2})^T \nabla_{\boldsymbol{w}_t} \mathcal{T}_{\boldsymbol{w}_t}(\boldsymbol{b}_{t-1})$ ;
$\quad\quad$ **end**
$\quad\quad$ $\boldsymbol{w}_t = \boldsymbol{w}_t + \rho_t g_{\boldsymbol{w}_t}$ with step-size $\rho_t$
$\quad$ **end**
$\quad$ $\boldsymbol{\psi} = \boldsymbol{\psi} + \eta_t \nabla_{\boldsymbol{\psi}} f(\boldsymbol{b}_{1:T}; \boldsymbol{\psi})$ with step-size $\eta_t$
**end**

---

## B  ORIGINAL DERIVATION OF THE AR AND ARM ESTIMATORS

Let us denote $t \sim \text{Exp}(\lambda)$ as an exponential distribution, whose probability density function is defined as $p(t \mid \lambda) = \lambda e^{-\lambda t}$, where $\lambda > 0$ and $t > 0$. The mean and variance are $\mathbb{E}[t] = \lambda^{-1}$ and $\text{var}[t] = \lambda^{-2}$, respectively. The exponential random variable $t \sim \text{Exp}(\lambda)$ can be reparameterized as

$t = \epsilon/\lambda$, $\epsilon \sim \text{Exp}(1)$. It is well known, *e.g.*, in Ross (2006), that if $t_1 \sim \text{Exp}(\lambda_1)$ and $t_2 \sim \text{Exp}(\lambda_2)$ are two independent exponential random variables, then the probability that $t_1$ is smaller than $t_2$ can be expressed as $P(t_1 < t_2) = \lambda_1/(\lambda_1 + \lambda_2)$; moreover, since $t_1 \sim \text{Exp}(\lambda_1)$ is equal in distribution to $\epsilon_1/\lambda_1$, $\epsilon_1 \sim \text{Exp}(1)$ and $t_2 \sim \text{Exp}(\lambda_2)$ is equal in distribution to $\epsilon_2/\lambda_2$, $\epsilon_2 \sim \text{Exp}(1)$, we have

$$P(t_1 < t_2) = P(\epsilon_1/\lambda_1 < \epsilon_2/\lambda_2) = P(\epsilon_1 < \epsilon_2\lambda_1/\lambda_2) = \lambda_1/(\lambda_1 + \lambda_2). \tag{27}$$

### B.1 AUGMENTATION OF A BERNOULLI RANDOM VARIABLE AND REPARAMETERIZATION

From (27) it becomes clear that the Bernoulli random variable $z \sim \text{Bernoulli}(\sigma(\phi))$ can be reparameterized by comparing two augmented exponential random variables as

$$z = \mathbf{1}_{[\epsilon_1 < \epsilon_2 e^\phi]}, \ \epsilon_1 \sim \text{Exp}(1), \ \epsilon_2 \sim \text{Exp}(1). \tag{28}$$

Consequently, the expectation with respect to the Bernoulli random variable can be reparameterized as one with respect to two augmented exponential random variables as

$$\mathcal{E}(\phi) = \mathbb{E}_{z \sim \text{Bernoulli}(\sigma(\phi))}[f(z)] = \mathbb{E}_{\epsilon_1, \epsilon_2 \overset{iid}{\sim} \text{Exp}(1)}[f(\mathbf{1}_{[\epsilon_1 e^{-\phi} < \epsilon_2]})]. \tag{29}$$

### B.2 REINFORCE ESTIMATOR IN THE AUGMENTED SPACE

Since the indicator function $\mathbf{1}_{[\epsilon_1 e^{-\phi} < \epsilon_2]}$ is not differentiable, the reparameterization trick in (2) is not directly applicable to computing the gradient of (29). Fortunately, as $t_1 = \epsilon_1 e^{-\phi}$, $\epsilon_1 \sim \text{Exp}(1)$ is equal in distribution to $t_1 \sim \text{Exp}(e^\phi)$, the expectation in (29) can be further reparameterized as

$$\mathcal{E}(\phi) = \mathbb{E}_{\epsilon_1, \epsilon_2 \overset{iid}{\sim} \text{Exp}(1)}[f(\mathbf{1}_{[\epsilon_1 e^{-\phi} < \epsilon_2]})] = \mathbb{E}_{t_1 \sim \text{Exp}(e^\phi), \ \epsilon_2 \sim \text{Exp}(1)}[f(\mathbf{1}_{[t_1 < \epsilon_2]})], \tag{30}$$

and hence, via REINFORCE and then another reparameterization, we can express the gradient as

$$\begin{aligned}
\nabla_\phi \mathcal{E}(\phi) &= \mathbb{E}_{t_1 \sim \text{Exp}(e^\phi), \ \epsilon_2 \sim \text{Exp}(1)}[f(\mathbf{1}_{[t_1 < \epsilon_2]})\nabla_\phi \log \text{Exp}(t_1; e^\phi)] \\
&= \mathbb{E}_{t_1 \sim \text{Exp}(e^\phi), \ \epsilon_2 \sim \text{Exp}(1)}[f(\mathbf{1}_{[t_1 < \epsilon_2]})(1 - t_1 e^\phi)] \\
&= \mathbb{E}_{\epsilon_1, \epsilon_2 \overset{iid}{\sim} \text{Exp}(1)}[f(\mathbf{1}_{[\epsilon_1 e^{-\phi} < \epsilon_2]})(1 - \epsilon_1)].
\end{aligned} \tag{31}$$

Similarly, we have $\mathcal{E}(\phi) = \mathbb{E}_{\epsilon_1, \epsilon_2 \overset{iid}{\sim} \text{Exp}(1)}[f(\mathbf{1}_{[\epsilon_1 < \epsilon_2 e^\phi]})] = \mathbb{E}_{\epsilon_1 \sim \text{Exp}(1), \ t_2 \sim \text{Exp}(-e^\phi)}[f(\mathbf{1}_{[\epsilon_1 < t_2]})]$, and hence can also express the gradient as

$$\begin{aligned}
\nabla_\phi \mathcal{E}(\phi) &= \mathbb{E}_{\epsilon_1 \sim \text{Exp}(1), \ t_2 \sim \text{Exp}(e^{-\phi})}[f(\mathbf{1}_{[\epsilon_1 < t_2]})\nabla_\phi \log \text{Exp}(t_2; e^{-\phi})] \\
&= -\mathbb{E}_{\epsilon_1 \sim \text{Exp}(1), \ t_2 \sim \text{Exp}(e^{-\phi})}[f(\mathbf{1}_{[\epsilon_1 < t_2]})(1 - t_2 e^{-\phi})] \\
&= -\mathbb{E}_{\epsilon_1, \epsilon_2 \overset{iid}{\sim} \text{Exp}(1)}[f(\mathbf{1}_{[\epsilon_1 e^{-\phi} < \epsilon_2]})(1 - \epsilon_2)].
\end{aligned} \tag{32}$$

Note that letting $\epsilon_1, \epsilon_2 \overset{iid}{\sim} \text{Exp}(1)$ is the same in distribution as letting

$$\epsilon_1 = \epsilon u, \ \epsilon_2 = \epsilon(1 - u), \quad \text{where } u \sim \text{Uniform}(0, 1), \ \epsilon \sim \text{Gamma}(2, 1), \tag{33}$$

which can be proved using $\text{Exp}(1) \overset{d}{=} \text{Gamma}(1, 1)$, $(u, 1 - u)^T \overset{d}{=} \text{Dirichlet}(\mathbf{1}_2)$, where $u \sim \text{Uniform}(0, 1)\}$, together with Lemma IV.3 of Zhou & Carin (2012); we use "$\overset{d}{=}$" to denote "equal in distribution." Thus, (B.2) can be reparameterized as

$$\nabla_\phi \mathcal{E}(\phi) = \mathbb{E}_{u \sim \text{Uniform}(0,1), \ \epsilon \sim \text{Gamma}(2,1)} \left[ f(\mathbf{1}_{[u < \sigma(\phi)]})(1 - \epsilon u) \right],$$

Applying Rao-Blackwellization (Casella & Robert, 1996), we can further express the gradient as

$$\nabla_\phi \mathcal{E}(\phi) = \mathbb{E}_{u \sim \text{Uniform}(0,1)} \left[ f(\mathbf{1}_{[u < \sigma(\phi)]})(1 - 2u) \right]. \tag{34}$$

Therefore, the gradient estimator shown above, the same as (6), is referred to as the Augment-REINFORCE (AR) estimator.

### B.3  MERGE OF REINFORCE GRADIENTS

A key observation of the paper is that by swapping the indices of the two $iid$ standard exponential random variables in (32), the gradient $\nabla_\phi \mathcal{E}(\phi)$ can be equivalently expressed as

$$\nabla_\phi \mathcal{E}(\phi) = -\mathbb{E}_{\epsilon_1, \epsilon_2 \overset{iid}{\sim} \text{Exp}(1)}[f(\mathbf{1}_{[\epsilon_2 e^{-\phi} < \epsilon_1]})(1 - \epsilon_1)]. \tag{35}$$

As the term inside the expectation in (31) and that in (35) could be highly positively correlated, we are motivated to merge (31) and (35) by sharing the same set of standard exponential random variables for Monte Carlo integration, which provides a new opportunity to well control the estimation variance (Owen, 2013). More specifically, simply taking the average of (31) and (35) leads to

$$\nabla_\phi \mathcal{E}(\phi) = \mathbb{E}_{\epsilon_1, \epsilon_2 \overset{iid}{\sim} \text{Exp}(1)} \left[ \left( f(\mathbf{1}_{[\epsilon_1 e^{-\phi} < \epsilon_2]}) - f(\mathbf{1}_{[\epsilon_2 e^{-\phi} < \epsilon_1]}) \right)(1/2 - \epsilon_1/2) \right]. \tag{36}$$

Note one may also take a weighted average of (31) and (35), and optimize the combination weight to potentially further reduce the variance of the estimator. We leave that for future study.

Note that (36) can be reparameterized as

$$\nabla_\phi \mathcal{E}(\phi) = \mathbb{E}_{u \sim \text{Uniform}(0,1), \, \epsilon \sim \text{Gamma}(2,1)} \left[ \left( f(\mathbf{1}_{[u > \sigma(-\phi)]}) - f(\mathbf{1}_{[u < \sigma(\phi)]}) \right)(\epsilon u/2 - 1/2) \right],$$

Applying Rao-Blackwellization (Casella & Robert, 1996), we can further express the gradient as

$$\nabla_\phi \mathcal{E}(\phi) = \mathbb{E}_{u \sim \text{Uniform}(0,1)} \left[ \left( f(\mathbf{1}_{[u > \sigma(-\phi)]}) - f(\mathbf{1}_{[u < \sigma(\phi)]}) \right)(u - 1/2) \right]. \tag{37}$$

Therefore, the gradient estimator shown above, the same as (7), is referred to as the Augment-REINFORCE-merge (ARM) estimator.

## C  PROOFS

*Proof of Proposition 2.* Since the gradients $g_{\text{ARM}}(u, \phi)$, $g_{\text{AR}}(u, \phi)$, and $g_{\text{R}}(z, \phi)$ are all unbiased, their expectations are the same as the true gradient $g_{true}(\phi) = \sigma(\phi)(1 - \sigma(\phi))[f(1) - f(0)]$. Denote $f_\Delta(u, \phi) = f(\mathbf{1}_{[u > \sigma(-\phi)]}) - f(\mathbf{1}_{[u < \sigma(\phi)]})$. Since

$$f_\Delta(u, \phi) = \begin{cases} 0, & \text{if } \sigma(-|\phi|) < u < \sigma(|\phi|), \\ f(1) - f(0), & \text{if } u > \sigma(|\phi|), \\ f(0) - f(1), & \text{if } u < \sigma(-|\phi|), \end{cases} \tag{38}$$

The second moment of $g_{\text{ARM}}(u, \phi)$ can be expressed as

$$\mathbb{E}_{u \sim \text{Uniform}(0,1)}[g_{\text{ARM}}^2(u, \phi)] = \mathbb{E}_{u \sim \text{Uniform}(0,1)}[f_\Delta^2(u, \phi)(u - 1/2)^2]$$
$$= \int_{\sigma(|\phi|)}^1 [f(1) - f(0)]^2 (u - 1/2)^2 du + \int_0^{\sigma(-|\phi|)} [f(0) - f(1)]^2 (u - 1/2)^2 du$$
$$= \frac{1}{12}[1 - (\sigma(|\phi|) - \sigma(-|\phi|))^3][f(1) - f(0)]^2$$

Denoting $t = \sigma(|\phi|) - \sigma(-|\phi|)$, we can re-express $g_{true}(\phi) = \frac{1}{4}(1 - t^2)[f(1) - f(0)]$. Thus, the variance of $g_{\text{ARM}}(u, \phi)$ can be expressed as

$$\begin{aligned} \text{var}[g_{\text{ARM}}(u, \phi)] &= \frac{1}{4} \left[ \frac{1}{3}(1 - t^3) - \frac{1}{4}(1 - t^2)^2 \right][f(1) - f(0)]^2 \\ &= \frac{1}{16}(1 - t)(t^3 + \frac{7}{3}t^2 + \frac{1}{3}t + \frac{1}{3})[f(1) - f(0)]^2 \\ &\leq \frac{1}{25}[f(1) - f(0)]^2, \end{aligned} \tag{39}$$

which reaches its maximum at $0.039788[f(1) - f(0)]^2$ when $t = \frac{\sqrt{5} - 1}{2}$.

For the REINFORCE gradient, we have

$$\begin{aligned} \mathbb{E}_{z \sim \text{Bernoulli}(\sigma(\phi))}[g_{\text{R}}^2(z, \phi)] &= \mathbb{E}_{z \sim \text{Bernoulli}(\sigma(\phi))} \left[ f^2(z)(z(1 - \sigma(\phi)) - \sigma(\phi)(1 - z))^2 \right] \\ &= \sigma(\phi)(1 - \sigma(\phi))[(1 - \sigma(\phi))f^2(1) + \sigma(\phi)f^2(0)]. \end{aligned}$$

Therefore the variance can be expressed as

$$\text{var}[g_{\text{R}}(u, \phi)]$$
$$=\sigma(\phi)(1 - \sigma(\phi))\left[(1 - \sigma(\phi))f^2(1) + \sigma(\phi)f^2(0) - \sigma(\phi)(1 - \sigma(\phi))[f(1) - f(0)]^2\right]$$
$$=\sigma(\phi)(1 - \sigma(\phi))[(1 - \sigma(\phi))f(1) + \sigma(\phi)f(0)]^2.$$

The largest variance satisfies

$$\sup_{\phi} \text{var}[g_{\text{R}}(z, \phi)] \geq \text{var}[g_{\text{R}}(z, 0)] = \frac{1}{16}[f(1) + f(0)]^2,$$

and hence when $f$ is always positive or negative, we have

$$\frac{\sup_{\phi} \text{var}[g_{\text{ARM}}(z, \phi)]}{\sup_{\phi} \text{var}[g_{\text{R}}(z, \phi)]} \leq \frac{16}{25}\left(1 - 2\frac{f(0)}{f(0) + f(1)}\right)^2 \leq \frac{16}{25}.$$

In summary, the ARM gradient has a variance that is bounded by $\frac{1}{25}(f(1) - f(0))^2$, and its worst-case variance is smaller than that of REINFORCE. $\qquad\square$

*Proof of Proposition 3.* We only need to prove for $K = 1$ and the proof for $K > 1$ automatically follows. Since $\mathbb{E}_{\boldsymbol{u}}[f(\mathbf{1}_{[\boldsymbol{u}<\sigma(\phi)]})^2(u_v - 1/2)^2] = \mathbb{E}_{\boldsymbol{u}}[f(\mathbf{1}_{[\boldsymbol{u}>\sigma(-\phi)]})^2(u_v - 1/2)^2]$, we have

$$\text{var}(g_{\text{ARM}_1, v}) - \text{var}(g_{\text{AR}_1, v}) = -3\mathbb{E}_{\boldsymbol{u}}[f(\mathbf{1}_{[\boldsymbol{u}<\sigma(\phi)]})^2(u_v - 1/2)^2] + \mathbb{E}_{\boldsymbol{u}}[f(\mathbf{1}_{[\boldsymbol{u}>\sigma(-\phi)]})^2(u_v - 1/2)^2]$$
$$- 2\mathbb{E}_{\boldsymbol{u}}[f(\mathbf{1}_{[\boldsymbol{u}>\sigma(-\phi)]})f(\mathbf{1}_{[\boldsymbol{u}<\sigma(\phi)]})(u_v - 1/2)^2]$$
$$= -\mathbb{E}_{\boldsymbol{u}}[f(\mathbf{1}_{[\boldsymbol{u}<\sigma(\phi)]})^2(u_v - 1/2)^2] - \mathbb{E}_{\boldsymbol{u}}[f(\mathbf{1}_{[\boldsymbol{u}>\sigma(-\phi)]})^2(u_v - 1/2)^2]$$
$$- 2\mathbb{E}_{\boldsymbol{u}}[f(\mathbf{1}_{[\boldsymbol{u}>\sigma(-\phi)]})f(\mathbf{1}_{[\boldsymbol{u}<\sigma(\phi)]})(u_v - 1/2)^2]$$
$$= -\mathbb{E}_{\boldsymbol{u}}\left[\left(f(\mathbf{1}_{[\boldsymbol{u}>\sigma(-\phi)]}) + f(\mathbf{1}_{[\boldsymbol{u}<\sigma(\phi)]})\right)^2(u_v - 1/2)^2\right]$$
$$\leq 0,$$

which shows that the estimation variance of $g_{\text{ARM}_K, v}$ is guaranteed to be lower than that of the $g_{\text{AR}_K, v}$, unless $f(\mathbf{1}_{[\boldsymbol{u}>\sigma(-\phi)]}) + f(\mathbf{1}_{[\boldsymbol{u}<\sigma(\phi)]}) = 0$ almost surely. Furthermore, since

$$\text{var}(g_{\text{ARM}_1, v}) - \text{var}(g_{\text{AR}_2, v}) = \mathbb{E}_{\boldsymbol{u}}[(f(\mathbf{1}_{[\boldsymbol{u}<\sigma(\phi)]}) - f(\mathbf{1}_{[\boldsymbol{u}>\sigma(-\phi)]}))^2(u_v - 1/2)^2]$$
$$- \mathbb{E}_{\boldsymbol{u}^{(1)},\boldsymbol{u}^{(2)}}[(f(\mathbf{1}_{[\boldsymbol{u}^{(1)}<\sigma(\phi)]})(u_v^{(1)} - 1/2) + f(\mathbf{1}_{[\boldsymbol{u}^{(2)}<\sigma(\phi)]})(u_v^{(2)} - 1/2))^2]$$
$$= - 2\mathbb{E}_{\boldsymbol{u}^{(1)}}[f(\mathbf{1}_{[\boldsymbol{u}^{(1)}<\sigma(\phi)]})(u_v^{(1)} - 1/2)]\mathbb{E}_{\boldsymbol{u}^{(2)}}[f(\mathbf{1}_{[\boldsymbol{u}^{(2)}<\sigma(\phi)]})(u_v^{(2)} - 1/2))]$$
$$- 2\mathbb{E}_{\boldsymbol{u}}[f(\mathbf{1}_{[\boldsymbol{u}<\sigma(\phi)]})f(\mathbf{1}_{[\boldsymbol{u}>\sigma(-\phi)]}))(u_v - 1/2)^2]$$
$$= - 2\left(\mathbb{E}_{\boldsymbol{u}}[f(\mathbf{1}_{[\boldsymbol{u}<\sigma(\phi)]})(u_v - 1/2)]\right)^2$$
$$- 2\mathbb{E}_{\boldsymbol{u}}[f(\mathbf{1}_{[\boldsymbol{u}<\sigma(\phi)]})f(\mathbf{1}_{[\boldsymbol{u}>\sigma(-\phi)]}))(u_v - 1/2)^2],$$

when $f$ is always positive or negative, the variance of $g_{\text{ARM}_K, v}$ is lower than that of $g_{\text{AR}_{2K}, v}$. $\qquad\square$

*Proof of Proposition 4.* Denoting $g(\boldsymbol{u}) = g_{\text{AR}}(\boldsymbol{u}) - \boldsymbol{b}(\boldsymbol{u})$, we have

$$\text{var}[g_v(\boldsymbol{u})] - \text{var}[g_{\text{AR}, v}(\boldsymbol{u})] = -2\mathbb{E}_{\boldsymbol{u}}[g_{\text{AR}, v}(\boldsymbol{u})b_v(\boldsymbol{u})] + \mathbb{E}_{\boldsymbol{u}}[b_v^2(\boldsymbol{u})].$$

To maximize the variance reduction, it is equivalant to consider the constrained optimization problem

$$\min_{b_v(\boldsymbol{u})} \; - 2\mathbb{E}_{\boldsymbol{u}}[g_{\text{AR}, v}(\boldsymbol{u})b_v(\boldsymbol{u})] + \mathbb{E}_{\boldsymbol{u}}[b_v^2(\boldsymbol{u})]$$
$$\text{subject to: } b_v(\boldsymbol{u}) = -b_v(1 - \boldsymbol{u}),$$

which is the same as a Lagrangian problem as

$$\min_{b_v(\boldsymbol{u}),\lambda_v(\boldsymbol{u})} \mathcal{L}(b_v(\boldsymbol{u}), \lambda_v(\boldsymbol{u})) = -2\mathbb{E}_{\boldsymbol{u}}[g_{\text{AR}, v}(\boldsymbol{u})b_v(\boldsymbol{u})] + \mathbb{E}_{\boldsymbol{u}}[b_v^2(\boldsymbol{u})] + \int \lambda_v(\boldsymbol{u})(b_v(\boldsymbol{u}) + b_v(1 - \boldsymbol{u}))d\boldsymbol{u}.$$

Setting $\frac{\delta \mathcal{L}}{\delta \lambda_v} = 0$ gives $b_v(\boldsymbol{u}) + b_v(1 - \boldsymbol{u}) = 0$. By writing $\int \lambda_v(\boldsymbol{u})(b_v(\boldsymbol{u}) + b_v(1 - \boldsymbol{u}))d\boldsymbol{u} = \int (\lambda_v(\boldsymbol{u}) + \lambda_v(1 - \boldsymbol{u}))b_v(\boldsymbol{u})d\boldsymbol{u}$ and setting $\frac{\delta \mathcal{L}}{\delta b_v} = 0$, we have

$$[2g_{\text{AR},v}(\boldsymbol{u}) - 2b_v(\boldsymbol{u})]p(\boldsymbol{u}) = \lambda_v(\boldsymbol{u}) + \lambda_v(1 - \boldsymbol{u}). \tag{40}$$

Interchange $\boldsymbol{u}$ and $1 - \boldsymbol{u}$ gives

$$[2g_{\text{AR},v}(1 - \boldsymbol{u}) - 2b_v(1 - \boldsymbol{u})]p(1 - \boldsymbol{u}) = \lambda_v(1 - \boldsymbol{u}) + \lambda_v(\boldsymbol{u}). \tag{41}$$

Solving (40) and (41) with $b_v(\boldsymbol{u}) + b_v(1 - \boldsymbol{u}) = 0$ and $p(\boldsymbol{u}) = p(1 - \boldsymbol{u})$, we have the optimal baseline function as $b_v^*(\boldsymbol{u}) = \frac{1}{2}(g_{\text{AR},v}(\boldsymbol{u}) - g_{\text{AR},v}(1 - \boldsymbol{u}))$. The proof is completed by noticing that $g_{\text{AR}}(\boldsymbol{u}) - b^*(\boldsymbol{u})$ is the same as the single sample gradient estimate under the ARM estimator. $\square$

*Proof of Corollary 5.* Since $b_v(\boldsymbol{u}) = c_v(1 - 2u)$ satisfies the anti-symmetric property, we can directly arrive at Corollary 5 using Proposition 4. Alternatively, since $\mathbb{E}_{\boldsymbol{u}}[f(\mathbf{1}_{[\boldsymbol{u}<\sigma(\boldsymbol{\phi})]})^2(u_v - 1/2)^2] = \mathbb{E}_{\boldsymbol{u}}[f(\mathbf{1}_{[\boldsymbol{u}>\sigma(-\boldsymbol{\phi})]})^2(u_v - 1/2)^2]$ and $\mathbb{E}_{\boldsymbol{u}}[f(\mathbf{1}_{[\boldsymbol{u}<\sigma(\boldsymbol{\phi})]})(u_v - 1/2)^2] = \mathbb{E}_{\boldsymbol{u}}[f(\mathbf{1}_{[\boldsymbol{u}>\sigma(-\boldsymbol{\phi})]})(u_v - 1/2)^2]$, for $g_{C,v} = (f(\mathbf{1}_{[\boldsymbol{u}<\sigma(\boldsymbol{\phi})]}) - c_v)(1 - 2u_v)$, we have

$$\text{var}(g_{C,v}) - \text{var}(g_{\text{ARM},v})$$
$$= \mathbb{E}_{\boldsymbol{u}}[(f(\mathbf{1}_{[\boldsymbol{u}<\sigma(\boldsymbol{\phi})]}) - c_v)^2(1 - 2u_v)^2] - \mathbb{E}_{\boldsymbol{u}}[(f(\mathbf{1}_{[\boldsymbol{u}<\sigma(\boldsymbol{\phi})]}) - f(\mathbf{1}_{[\boldsymbol{u}>\sigma(-\boldsymbol{\phi})]}))^2(u_v - 1/2)^2]$$
$$= \mathbb{E}_{\boldsymbol{u}}\left[\left(4c_v^2 - 8c_v f(\mathbf{1}_{[\boldsymbol{u}<\sigma(\boldsymbol{\phi})]}) + 2f^2(\mathbf{1}_{[\boldsymbol{u}<\sigma(\boldsymbol{\phi})]}) + 2f(\mathbf{1}_{[\boldsymbol{u}<\sigma(\boldsymbol{\phi})]})f(\mathbf{1}_{[\boldsymbol{u}>\sigma(-\boldsymbol{\phi})]})\right)(u_v - 1/2)^2\right]$$
$$= \mathbb{E}_{\boldsymbol{u}}\left[\left(f(\mathbf{1}_{[\boldsymbol{u}>\sigma(-\boldsymbol{\phi})]}) + f(\mathbf{1}_{[\boldsymbol{u}<\sigma(\boldsymbol{\phi})]}) - 2c_v\right)^2(u_v - 1/2)^2\right]$$
$$\geq 0.$$

$\square$

*Proof of Proposition 6.* First, to compute the gradient with respect to $\boldsymbol{w}_1$, since
$$\mathcal{E}(\boldsymbol{w}_{1:T}) = \mathbb{E}_{q(\boldsymbol{b}_1)}\mathbb{E}_{q(\boldsymbol{b}_{2:T} \mid \boldsymbol{b}_1)}[f(\boldsymbol{b}_{1:T})], \tag{42}$$
we have
$$\nabla_{\boldsymbol{w}_1}\mathcal{E}(\boldsymbol{w}_{1:T}) = \mathbb{E}_{\boldsymbol{u}_1 \sim \text{Uniform}(0,1)}[f_\Delta(\boldsymbol{u}_1, \mathcal{T}_{\boldsymbol{w}_1}(\boldsymbol{x}))(\boldsymbol{u}_1 - 1/2)]\nabla_{\boldsymbol{w}_1}\mathcal{T}_{\boldsymbol{w}_1}(\boldsymbol{x}), \tag{43}$$
where
$$f_\Delta(\boldsymbol{u}_1, \mathcal{T}_{\boldsymbol{w}_1}(\boldsymbol{x})) = \mathbb{E}_{\boldsymbol{b}_{2:T} \sim q(\boldsymbol{b}_{2:T} \mid \boldsymbol{b}_1),\, \boldsymbol{b}_1 = \mathbf{1}_{[\boldsymbol{u}_1 > \sigma(-\mathcal{T}_{\boldsymbol{w}_1}(\boldsymbol{x}))]}}[f(\boldsymbol{b}_{1:T})]$$
$$- \mathbb{E}_{\boldsymbol{b}_{2:T} \sim q(\boldsymbol{b}_{2:T} \mid \boldsymbol{b}_1),\, \boldsymbol{b}_1 = \mathbf{1}_{[\boldsymbol{u}_1 < \sigma(\mathcal{T}_{\boldsymbol{w}_1}(\boldsymbol{x}))]}}[f(\boldsymbol{b}_{1:T})]. \tag{44}$$

Second, to compute the gradient with respect to $\boldsymbol{w}_t$, where $2 \leq t \leq T - 1$, since
$$\mathcal{E}(\boldsymbol{w}_{1:T}) = \mathbb{E}_{q(\boldsymbol{b}_{1:t-1})}\mathbb{E}_{q(\boldsymbol{b}_t \mid \boldsymbol{b}_{t-1})}\mathbb{E}_{q(\boldsymbol{b}_{t+1:T} \mid \boldsymbol{b}_t)}[f(\boldsymbol{b}_{1:T})], \tag{45}$$
we have
$$\nabla_{\boldsymbol{w}_t}\mathcal{E}(\boldsymbol{w}_{1:T}) = \mathbb{E}_{q(\boldsymbol{b}_{1:t-1})}\left[\mathbb{E}_{\boldsymbol{u}_t \sim \text{Uniform}(0,1)}[f_\Delta(\boldsymbol{u}_t, \mathcal{T}_{\boldsymbol{w}_t}(\boldsymbol{b}_{t-1}), \boldsymbol{b}_{1:t-1})(\boldsymbol{u}_t - 1/2)]\nabla_{\boldsymbol{w}_t}\mathcal{T}_{\boldsymbol{w}_t}(\boldsymbol{b}_{t-1})\right], \tag{46}$$
where
$$f_\Delta(\boldsymbol{u}_t, \mathcal{T}_{\boldsymbol{w}_t}(\boldsymbol{b}_{t-1}), \boldsymbol{b}_{1:t-1}) = \mathbb{E}_{\boldsymbol{b}_{t+1:T} \sim q(\boldsymbol{b}_{t+1:T} \mid \boldsymbol{b}_t),\, \boldsymbol{b}_t = \mathbf{1}_{[\boldsymbol{u}_t > \sigma(-\mathcal{T}_{\boldsymbol{w}_t}(\boldsymbol{b}_{t-1}))]}}[f(\boldsymbol{b}_{1:T})]$$
$$- \mathbb{E}_{\boldsymbol{b}_{t+1:T} \sim q(\boldsymbol{b}_{t+1:T} \mid \boldsymbol{b}_t),\, \boldsymbol{b}_t = \mathbf{1}_{[\boldsymbol{u}_t < \sigma(\mathcal{T}_{\boldsymbol{w}_t}(\boldsymbol{b}_{t-1}))]}}[f(\boldsymbol{b}_{1:T})]. \tag{47}$$

Finally, to compute the gradient with respect to $\boldsymbol{w}_T$, we have
$$\nabla_{\boldsymbol{w}_T}\mathcal{E}(\boldsymbol{w}_{1:T}) = \mathbb{E}_{q(\boldsymbol{b}_{1:T-1})}\left[\mathbb{E}_{\boldsymbol{u}_T \sim \text{Uniform}(0,1)}[f_\Delta(\boldsymbol{u}_T, \mathcal{T}_{\boldsymbol{w}_T}(\boldsymbol{b}_{T-1}), \boldsymbol{b}_{1:T-1})(\boldsymbol{u}_T - 1/2)] \right.$$
$$\left. \times \nabla_{\boldsymbol{w}_T}\mathcal{T}_{\boldsymbol{w}_T}(\boldsymbol{b}_{T-1})\right], \tag{48}$$
where
$$f_\Delta(\boldsymbol{u}_T, \mathcal{T}_{\boldsymbol{w}_T}(\boldsymbol{b}_{T-1}), \boldsymbol{b}_{1:T-1}) = f(\boldsymbol{b}_{1:T-1}, \boldsymbol{b}_T = \mathbf{1}_{[\boldsymbol{u}_T > \sigma(-\mathcal{T}_{\boldsymbol{w}_T}(\boldsymbol{b}_{T-1}))]})$$
$$- f(\boldsymbol{b}_{1:T-1}, \boldsymbol{b}_T = \mathbf{1}_{[\boldsymbol{u}_T < \sigma(\mathcal{T}_{\boldsymbol{w}_T}(\boldsymbol{b}_{T-1}))]}). \tag{49}$$

$\square$

## D   ADDITIONAL EXPERIMENTAL RESULTS

For the univariate AR gradient, we have

$$\mathbb{E}_{u \sim \text{Uniform}(0,1)}[g_{\text{AR}}^2(u, \phi)] = \int_0^{\sigma(\phi)} f^2(1)(1-2u)^2 du + \int_{\sigma(\phi)}^1 f^2(0)(1-2u)^2 du$$

$$= \frac{1}{6}(f^2(0) + f^2(1)) + \frac{1}{6}(1 - 2\sigma(\phi))^3(f^2(0) - f^2(1)).$$

Thus its variance can be expressed as

$$\text{var}[g_{\text{AR}}(u, \phi)] = \frac{1}{6}(f^2(0) + f^2(1)) + \frac{1}{6}(1 - 2\sigma(\phi))^3(f^2(0) - f^2(1)) - \sigma^2(\phi)(1 - \sigma(\phi))^2[f(1) - f(0)]^2,$$

which is used for related plots in Figures 1 and 5.

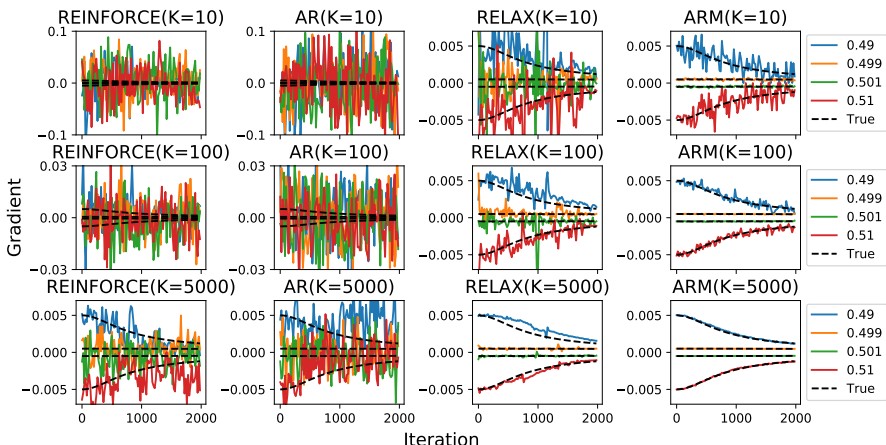

Figure 4: Comparison of a variety of gradient estimators for estimating the gradient of $\mathcal{E}(\phi) = \mathbb{E}_{z \sim \text{Bernoulli}(\sigma(\phi))}[(z - p_0)^2]$, where $p_0 \in \{0.49, 0.499, 0.501, 0.51\}$ and the values of $\phi$ are updated via gradient ascent with the true gradients. Shown in Rows 1, 2, and 3 are the trace plots of the estimated gradients using $K = 10$, 100, and 5000 Monte Carlo samples, respectively.

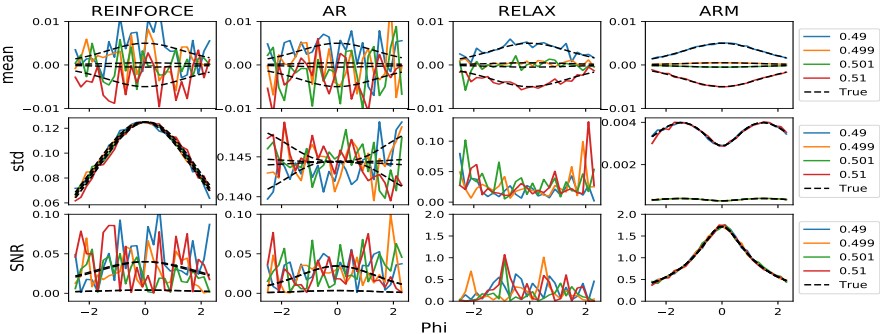

Figure 5: Comparison of a variety of gradient estimators for estimating the gradient of $\mathcal{E}(\phi) = \mathbb{E}_{z \sim \text{Bernoulli}(\sigma(\phi))}[(z - p_0)^2]$, where $p_0 \in \{0.49, 0.499, 0.501, 0.51\}$ and the values of $\phi$ range from $-2.5$ to 2.5. For each $\phi$ value, we compute for each estimator $K = 1000$ single-Monte-Carlo-sample gradient estimates, and use them to calculate their sample mean $\bar{g}$, sample standard deviation $s_g$, and gradient signal-to-noise ratio $\text{SNR}_g = |\bar{g}|/s_g$. In each estimator specific column, we plot $\bar{g}$, $s_g$, and $\text{SNR}_g$ in Rows 1, 2, and 3, respectively. The theoretical gradient standard deviations and gradient signal-to-noise ratios are also shown if they can be analytically calculated (see Eq. 25 and Appendices C and D and for related analytic expressions).

Table 4: Test negative ELBOs of discrete VAEs trained with four different stochastic gradient estimators. MNIST-threshold is the binarized MNIST thresholded at 0.5 and MNIST-static is the binarized MNIST used in Salakhutdinov & Murray (2008); Larochelle & Murray (2011).

|  |  |  | ARM | RELAX | REBAR | ST Gumbel-Softmax |
|---|---|---|---|---|---|---|
| Bernoulli | Nonlinear | MNIST-threshold | **101.3** | 110.9 | 111.6 | 112.5 |
|  |  | MNIST-static | **109.9** | 112.1 | 111.8 | - |
|  |  | OMNIGLOT | 129.5 | **128.2** | 128.3 | 140.7 |
|  | Linear | MNIST-threshold | **110.3** | 122.1 | 123.2 | 129.2 |
|  |  | MNIST-static | **116.2** | 116.7 | 117.9 | - |
|  |  | OMNIGLOT | **124.2** | 124.4 | 124.9 | 129.8 |
|  | Two layers | MNIST-threshold | **98.2** | 114.0 | 113.7 | - |
|  |  | MNIST-static | 105.8 | 105.6 | **105.5** | - |
|  |  | OMNIGLOT | **118.3** | 119.1 | 118.8 | - |

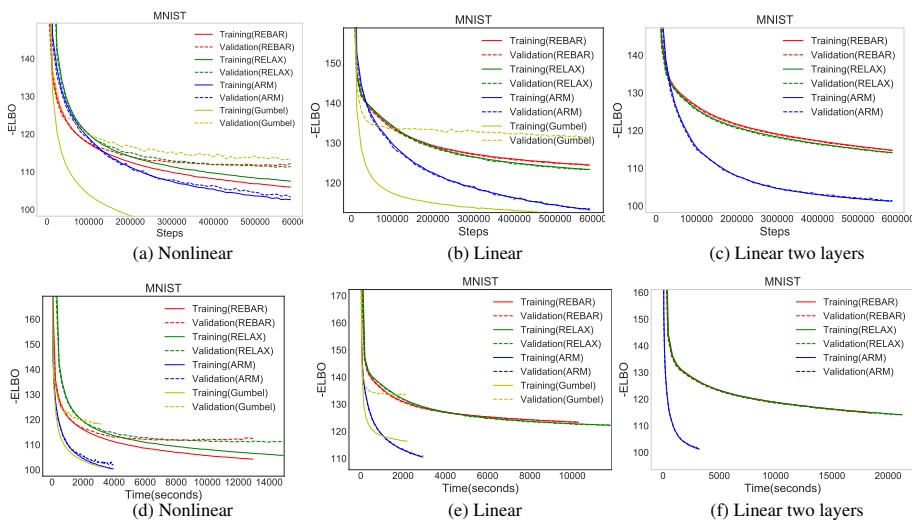

Figure 6: Training and validation negative ELBOs on MNIST-threshold with respect to the training iterations, shown in the top row, and with respect to the wall clock times on Tesla-K40 GPU, shown in the bottom row, for three differently structured Bernoulli VAEs.

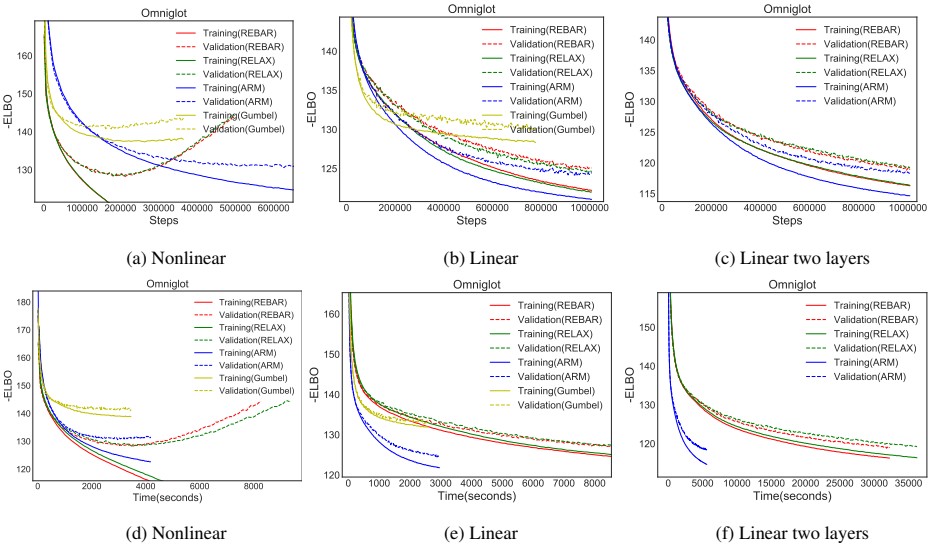

Figure 7: Training and validation negative ELBOs on OMNIGLOT with respect to the training iterations, shown in the top row, and with respect to the wall clock times on Tesla-K40 GPU, shown in the bottom row, for three differently structured Bernoulli VAEs.

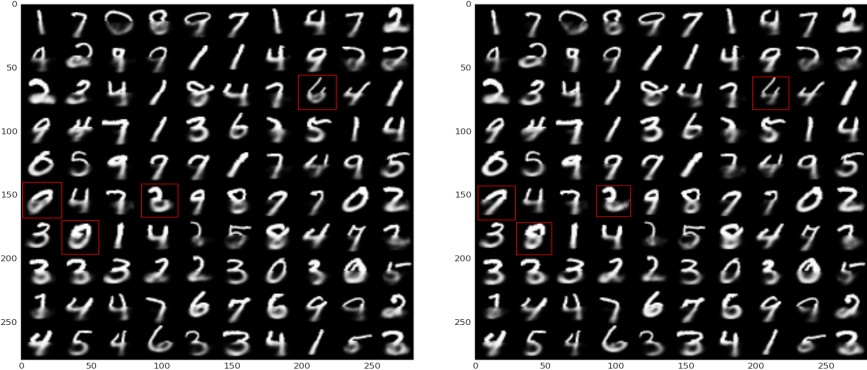

Figure 8: Randomly selected example results of predicting the lower half of a MNIST digit given its upper half, using a binary stochastic network, which has two binary linear stochastic hidden layers and is trained by the ARM estimator based maximum likelihood estimation. Red squares highlight notable variations between two random draws.

