# OpenReview forum: "ARM: Augment-REINFORCE-Merge Gradient for Stochastic Binary Networks"
_ICLR.cc/2019/Conference_

### Official Review · AnonReviewer1 · 2018-11-02
**This paper is very good but needs improvement**

**Rating:** 7
**Confidence:** 3

**Review:**

For binary layers, how to calculate and backpropagate gradients is a big problem, particularly for the binary neural networks. To solve the problem, this paper proposes an unbiased and low variance augment-REINFORCE-merge (ARM) estimator. With the help of an appropriate reparameterization, the antithetic sampling in an augmented space can be used to drive a variance-reduction mechanism. The experimental results show that ARM estimator converges fast, has low computational complexity, and provides advanced prediction performance.

This paper is well-organized. The motivation of the proposed model is well-driven and algorithm is articulated clearly. Meanwhile, the derivations and analysis of the proposed algorithm are correct. The experimental results show that the proposed model is better than the other existing methods.

A few minor revision are list below.
1) In figure 1, it seems difficult to decide which one is better from the trace plots of the true/estimated gradients. Also, why the author choose to compare the REINFORCE instead of REBAR and RELAX, since REBAR and RELAX improve on REINFORCE by introducing stochastically estimated control variates. Also, about trace plots of the loss functions, I am curious why REINFORCE has a big vibration during 1500~2000 iterations.
2) About Table 2, are all compared methods in the same experimental settings?

---

> ### Author Response · Authors · 2018-11-06
> **We are making all the suggested minor revisions**
>
> Thank you very much for your positive feedback. We have made all these suggested minor revisions in the updated paper. Please see our point-by-point response to your comments below.
>
> Q1) 1) In figure 1, it seems difficult to decide which one is better from the trace plots of the true/estimated gradients.
>
> A1: For this univariate binary toy example, the trace plots of the true gradients of $\phi$, shown in the first subplot of the top row, seem very different from these of the estimated gradients with ARM, shown in the last subplot of the top row. However, the trace plots of the Bernoulli probability parameter $\sigma(\phi)$ updated with the true gradient, shown in the first subplot of the second row, are almost indistinguishable from these updated with the ARM gradient, shown in the last subplot of the second row. Given the same step-size of one, it is hard to tell whether the ARM or true gradient is better for updating $\phi$, which is showing how surprisingly well ARM works!
>
> Q2: Also, why the author choose to compare the REINFORCE instead of REBAR and RELAX, since REBAR and RELAX improve on REINFORCE by introducing stochastically estimated control variates.
>
> A2: In the revised paper, we have revised Figure 1 to add the trace plots of the estimated gradients, Bernoulli probability parameters, and empirical gradient variance for five different algorithms: True grad, REINFORCE, Augment-REINFORCE, RELAX, and ARM. We have also added Figures 4 and 5 to provide more information.
>
> Q3: Also, about trace plots of the loss functions, I am curious why REINFORCE has a big vibration during 1500~2000 iterations.
>
> A3: The REINFORCE has large variance, which sometimes leads to divergence. In the revised paper, we have tried a much reduced step-size for REINFORCE and updated the figures accordingly. We find that the volatility can be clearly reduced but the objective could sometime still converge to the wrong point as the learning progresses.
>
> If you are interested in playing with this toy examples by yourself, you may run ARM_toy.py provided in the anonymous Github code repository.
>
> Q4: 2) About Table 2, are all compared methods in the same experimental settings?
>
> A4: We tried our best to make a comparison that is as fair as possible: 1) We have ensured that we are using the same version of banarized MNIST for all algorithms. 2) We have ensured all methods use the same network size; if the original paper used different ones, we have modified and run the author provided code (eg. LeGrad) to ensure the comparability.  3) We have run five independent trials to add error bars to make the comparison more meaningful (many previous work did not report error bars).

---

### Official Review · AnonReviewer2 · 2018-11-02
**REVISED: ARM algorithm is an interesting approach to a limited domain of interest in ML. While limited, it may spark new research into augmentation of random variables for variance reduction**

**Rating:** 6
**Confidence:** 4

**Review:**

Overview.
The authors present an algorithm for lowering the variance of the score-function gradient estimator in the special case of stochastic binary networks. The algorithm, called Augment-REINFORCE-merge proceeds by augmenting binary random variables. ARM combines Rao-Blackwellization and common random numbers (equivalent to antithetic sampling in this case, due to symmetry) to produce what the authors claim to be a lower variance gradient estimator. The approach is somewhat novel. I have not seen other authors attempt to apply REINFORCE in an augmented space and with antithetic samples / common random numbers, and Rao-Blackwellization. This combination of techniques may be a good idea in the case of Bernoulli random variables. However, due to a number of issues discussed below, this claim is not possible to evaluate from the paper.

Issues/Concerns
- I assess the paper in its current form as too far below the acceptable standard in writing and in clarity of presentation, setting aside other conceptual issues which I discuss below. The paper contains many typos and a few run-on sentences that span 5-7 lines. This hinders understanding substantially. A number key terms are not explained, irregularly. Although the paper assumes that readers do not know the mean and a variance of a Bernoulli random variable, or theof  definition of an indicator function, it does not explain what random variable augmentation means. The one sentence that comes close to explaining it seems to have a typo: "From (5) it becomes clear that the Bernoulli random variable z ∼ Bernoulli(σ(φ)) can be reparameterized by racing two augmented exponential random variables ...". It is not clear what is meant by "racing," here, and I do not find it clear from equation (5) what is going on. Unfortunately, in the abstract, the paper claims that variance reduction is achieved by "data augmentation," which has a very specific meaning in machine learning unrelated to augmented random variables, further obfuscating meaning. Similarly, the term "merge" is not explained, despite the subheading 2.3.
- Computational issues are not addressed in the paper. Whether or not this method is useful in practice depends on computational complexity
- No effort is made to diagnose the source of the variance reduction, other than in the special case of analytically comparing with the Augment-REINFORCE estimator, which does not appear in any of the experiments.
- No effort is made to empirically characterize the variance of the gradient estimator, unlike Tucker et al (2017) and Grathwohl et al. (2018).
- The algorithm presented in the appendix appears to only address single-layer stochastic binary networks, which are uninteresting in practice.
- Figure 2 (d), (e), and (f) all show that ARM was stopped early. Given that RELAX and REBAR overfit, this is a little troubling. Overal, these results are not very convincing that ARM is better, particularly in the absence of variance analysis (empirically, or other than w.r.t. the same algorithm without the merge step). All algorithms should be run for the same number of steps, particularly in cases where they may be prone to overfitting.
- Figure 1 I believe contains an error for the REINFORCE figure. In my own research I have run these experiments myself, with a value of p close to the one used by the authors. REBAR and RELAX both reduce to a REINFORCE gradient estimator with a control variate that is differentiably reparametrizable, and so the erratic behaviour of the REINFORCE estimator in this case is likely wrong.
- There is a mysterious sentence on page 6 that refers to ARM adjusting the "frequencies, amplitudes, and signs of its gradient estimates with larger and more frequent spikes for larger true gradients"
-The value to the community of another gradient estimator for binary random variables is low, given the plethora of other methods available. Given the questions remaining about this methodology and its experiments, I recommend against publication on this basis also.
- Table 2 compares results that mix widely different architectures against each other, some taken directly from papers, others possibly retrained. This is not a valid comparison to make when evaluating a new gradient estimator, where the model must be fixed.


* EDIT: I have re-evaluated the careful and comprehensive response to my concerns by the authors. I thank them for their effort in this. As many of the concerns were related to communication and have been addressed in the most recent draft, I think it is appropriate to move my review upwards. The revisions make this paper quite different from the original, and I am happy to re-evaluate on that basis--this is a peculiarity of the ICLR open review procedure, but I consider it a strength.

I note that "data augmentation" in machine learning appears to have collided with a term in the Bayesian statistics literature, and the authors have provide a number of citations to support this. I strongly recommend "variable augmentation" going forward, as that is an accurate description (you are augmenting a random variable, rather than the input data domain). This appears to be one of the growing pains of the field of ML which has distinct and often orthogonal concerns to classical statistics around density approximation and computational issues.*

---

> ### Author Response · Authors · 2018-11-06
> **Point-by-point response to address the raised issues/concerns of AR2**
>
>
> Comments: The authors present ... The approach is somewhat novel. I have not seen other authors attempt to apply REINFORCE in an augmented space and with antithetic samples / common random numbers, and Rao-Blackwellization. This combination of techniques may be a good idea in the case of Bernoulli random variables. However, due to a number of issues discussed below, this claim is not possible to evaluate from the paper.
>
> Response: We thank reviewer 2 for his/her detailed comments. It appears that the reviewer has a good understanding about the technical novelty of the paper, but is not convinced by the claim of the paper. Below please find our point-by-point response, which we believe will be able to address all the raised issues/concerns.
>
> Q1) I assess the paper in its current form as too far below the acceptable standard in writing and in clarity of presentation, setting aside other conceptual issues which I discuss below.
>
> A1: We have tried very hard to make the paper easy to follow. We have further simplified the derivation of ARM gradient in the revised version with the longer original derivation deferred to the Appendix B.
>
> Q2) The paper contains many typos and a few run-on sentences that span 5-7 lines. This hinders understanding substantially.
>
> A2: We sincerely apologize for possible typos and we appreciate if Reviewer 2 could help point them out. We have rewritten several long sentences into shorter ones.
>
> Q3) A number key terms are not explained, irregularly. Although the paper assumes that readers do not know the mean and a variance of a Bernoulli random variable, or theof  definition of an indicator function, it does not explain what random variable augmentation means.
>
> A3: We consider that variable augmentation is a well-known concept to readers familiar with statistical models and inference algorithms (such as the EM algorithm). In addition, we thought Eq (6) (Eq 27 of the revised paper) is self-explanatory given the provided background information about exponential random variables and Eq (5) (Eq 26 of the revised paper).
>
> In the revision, we have added citations to classical papers on variable augmentation. Moreover, our derivation of the univariate ARM estimator, shown in Section 2.1 of the revised paper, is now significantly simplified and no longer relies on variable augmentation.
>
> Q4) The one sentence that comes close to explaining it seems to have a typo: "From (5) it becomes clear that the Bernoulli random variable z ~ Bernoulli(σ(φ)) can be reparameterized by racing two augmented exponential random variables ...". It is not clear what is meant by "racing," here, and I do not find it clear from equation (5) what is going on.
>
> A4:  "Racing," which is related to the well-known "Exponential Race Problem," is not a typo. We apologize if we did not make the analogy between "racing two exponential random variables" and "treating the smaller one of two exponential random variables as the winner" clear. We have changed "racing" to "comparing" in the revised paper.
>
> Q5) Unfortunately, in the abstract, the paper claims that variance reduction is achieved by "data augmentation," which has a very specific meaning in machine learning unrelated to augmented random variables, further obfuscating meaning.
>
> A5: As a compromise, we have changed "data augmentation" to "variable augmentation" to reduce confusion. The reason we used "data augmentation" was because it is very widely used in both statistics and machine learning literature. Its origin is often attributed to the following highly cited paper:
>
> M. A. Tanner and W. H. Wong, The Calculation of Posterior Distributions by Data Augmentation (Discussion Article), Journal of the American Statistical Association, June 1987.
>
> Below please find several additional examples that can help justify our use of "data augmentation:"
>
> D. A. van Dyk and X.-L. Meng, The Art of Data Augmentation (Discussion Article), Journal of Computational and Graphical Statistics, Mar., 2001.
>
> M. A. Tanner and W. H. Wong, From EM to Data Augmentation: The Emergence of MCMC Bayesian Computation in the 1980s, Statistical Science, 2010.
>
> N. G. Polson and S. L. Scott, Data Augmentation for Support Vector Machines, Bayesian Analysis, 2011.
>
> K. P. Murphy, Machine Learning: A Probabilistic Perspective (Chapter 24.2.7, Page 847), 2012.
>
> M. Xu, J. Zhu, and B. Zhang, Fast Max-Margin Matrix Factorization with Data Augmentation, ICML 2013.
>
> Z. Gan, R. Henao, D. Carlson, and L. Carin, Learning Deep Sigmoid Belief Networks with Data Augmentation, AISTATS 2015.
>
> Again, we have changed "data augmentation" to "variable augmentation" to avoid possible confusions. We have also added two classical references for this concept.

---

> > ### Author Response · Authors · 2018-11-06
> > **Point-by-point response, part 2**
> >
> >
> > Q6) Similarly, the term "merge" is not explained, despite the subheading 2.3.
> >
> > A6: We thought we had clearly defined "merge" as "sharing the same set of standard exponential random variables for Monte Carlo integration, ..., More specifically, simply taking the average of (9) and (11) leads to (12)." Our new simplified derivation no longer requires this "merge" step, which is now deferred to the Appendix as part of the original derivation for ARM.
> >
> > Q7) Computational issues are not addressed in the paper. Whether or not this method is useful in practice depends on computational complexity
> >
> > A7: We totally agree that "Whether or not this method is useful in practice depends on computational complexity" but we respectively disagree "Computational issues are not addressed in the paper." We'd like to emphasize that in Figures 2, 5, 6 (Figures 2, 6, 7 of the revised paper), we plot the calculated training and validation ELBOs against the number of processed mini-batches (steps) in the first row, and replot the same ELBOs against the computational time in the second row. These Figures suggest ARM takes clearly shorter time to finish the same (or more) number of iterations. We have added more explanations to these Figures in our revision.
> >
> > Q8) No effort is made to diagnose the source of the variance reduction, other than in the special case of analytically comparing with the Augment-REINFORCE estimator, which does not appear in any of the experiments.
> >
> > A8: This is a good point. We have added theoretical variance reduction of the ARM gradient estimator over REINFORCE and Augment-REINFORCE (AR) estimators in Section 2.3 of the revised version. The newly added Proposition 2 compares ARM with REINFORCE, Propositions 3-4 compare ARM with AR, and Corollary 5 compares ARM with a constant based baseline. The -log p(x) for AR on MNIST are 164.1, 114.6, and 162.2 for the “Linear,” “Nonlinear,” and “Two layers” networks, respectively, which are comparable to these of REINFORCE. We have added them to Table 2.
> >
> > Q9) No effort is made to empirically characterize the variance of the gradient estimator, unlike Tucker et al (2017) and Grathwohl et al. (2018).
> >
> > A9: This is a good point. We have added empirical variance plots in Figures 1, 3, 5, following Tucker et al (2017) and Grathwohl et al. (2018).
> >
> > Q10) The algorithm presented in the appendix appears to only address single-layer stochastic binary networks, which are uninteresting in practice.
> >
> > A10: Algorithm 1 in Appendix is describing a generic ARM algorithm (please note we had the following clarification in Appendix A: "For stochastic transforms, the implementation of ARM gradient is discussed in Section 3."). Describing the ARM algorithm for a multi-stochastic-layer network is the sole purpose of Section 3. For a network with multiple stochastic hidden layers, the ARM algorithm is described in Proposition 2 (Proposition 6 of the revised paper)  if variational auto-encoder is used, and in Proposition 3 (Proposition 7 in the revised paper) if maximum likelihood is used. In the revision, we have added Algorithm 2 to further describe the ARM gradient for a multi-stochastic-layer network.
> >
> > Q11) Figure 2 (d), (e), and (f) all show that ARM was stopped early. Given that RELAX and REBAR overfit, this is a little troubling.
> >
> > A11: All methods in Figure 2 are compared by running the same number of iterations. If an algorithm is faster, it will take less time to complete the given number of iterations. In Figure 2 (d)-(f), ARM appeared to be stopped early only because it takes much less time than REBAR/RELAX does to finish the same number of iterations because it is much faster per iteration. We have revised the paper accordingly to enhance clarity. Please also see our response in A7.
> >
> > Q12) Overal, these results are not very convincing that ARM is better, particularly in the absence of variance analysis (empirically, or other than w.r.t. the same algorithm without the merge step).
> >
> > A12: As commented in A8 and A9, we have added variance analysis both empirically and theoretically in the revised version.
> >
> > Q13) All algorithms should be run for the same number of steps, particularly in cases where they may be prone to overfitting.
> >
> > A13: We totally agree with the comment and this was actually what we did, as shown in Figure 2 (a)-(c) (in fact, we had tried running ARM with more number of steps to see whether it would overfit eventually; we did not observe overfitting with more iterations). Please also see our response in A11.

---

> > > ### Author Response · Authors · 2018-11-06
> > > **Point-by-point response, part 3**
> > >
> > >
> > > Q14) Figure 1 I believe contains an error for the REINFORCE figure. In my own research I have run these experiments myself, with a value of p close to the one used by the authors. REBAR and RELAX both reduce to a REINFORCE gradient estimator with a control variate that is differentiably reparametrizable, and so the erratic behaviour of the REINFORCE estimator in this case is likely wrong.
> > >
> > > A14: REINFORCE without an appropriate control variate may have huge variance and hence has erratic behavior at certain iterations (e.g., wrong convergence point) even if the step-size is set to be small. Note we set the step-size as one in the original paper. We have now reduced the REINFORCE stepsize for the toy example to be 0.1, under which the parameter changes more smoothly and converges more slowly, but it sometimes still diverges due to high gradient variance.
> > >
> > > We appreciate if the reviewer could try our demo code "ARM_toy.py" in the provided anonymous GitHub code repository if he/she still believes there is an error for the REINFORCE figure.
> > >
> > > Q15) There is a mysterious sentence on page 6 that refers to ARM adjusting the "frequencies, amplitudes, and signs of its gradient estimates with larger and more frequent spikes for larger true gradients"
> > >
> > > A15: We have added more explanations about that sentence. Please see the first Paragraph in Page 7 for details.
> > >
> > > Q16) The value to the community of another gradient estimator for binary random variables is low, given the plethora of other methods available.
> > >
> > > A16: ARM runs much faster, is unbiased and has low variance, delivers similar or higher testing log-likelihood and ELBOs, and is almost as simple as REINFORCE to implement. We have added more discussions in the revised paper about how it is different from previously proposed ones. We believe its practical and theoretical value will be appreciated by the community and it can be potentially plugged into many other research tasks, such as the ones mentioned in Conclusions. In fact, we have recently found a paper submitted to ICLR this year that had independently verified the correctness and good performance of ARM in its experiments (to preserve anonymity, we cannot reveal the name of that paper).
> > >
> > > Q17) Given the questions remaining about this methodology and its experiments, I recommend against publication on this basis also.
> > >
> > > A17: We believe we have addressed all your questions, and hence we appreciate if you could take another look at our paper and reconsider your rating.
> > >
> > > Q18) Table 2 compares results that mix widely different architectures against each other, some taken directly from papers, others possibly retrained. This is not a valid comparison to make when evaluating a new gradient estimator, where the model must be fixed.
> > >
> > > A18: For the results taken from literature, we have tried our best to ensure that the models are the same as the ones we use, i.e., only the gradient estimator is different (we had communicated with some authors of these papers to double check); for the models which are different from the original papers, we modify the author provided code with the same network. All efforts had been made to ensure a fair and meaningful comparison.  Please also see our response A4 to Reviewer 1.

---

> ### Author Response · Authors · 2018-11-27
> **Thank you for re-evaluating our paper based on our revision and response**
>
> We greatly appreciate that you have taken our revision and response into consideration and moved your rating upwards.
>
> We agree with your suggestion on using "variable augmentation" when describing the augmentation of a random variable.

---

### Official Review · AnonReviewer3 · 2018-11-12
**An interesting paper with the potential to inspire many possible extensions, but overly complicated presentation (addressed in review).**

**Rating:** 8
**Confidence:** 4

**Review:**

In this paper the authors propose a new variance-reduction technique to use when computing an expected loss gradient where the expectation is with respect to independent binary random variables, e.g. for training VAEs with a discrete latent space. The paper is interesting, highly relevant, simple to implement, suggests many possible extensions, and shows good results on the experiments performed. However the exposition leaves a lot to be desired.

Major comments:

The authors devote several pages of fairly dense mathematics to deriving the ARM estimate in section 2 (up to section 2.5). However I found it relatively easy to derive (15) directly, using elementary results such as the law of total expectation and a single 1-dimensional integral, in about 10 lines of equations. As the authors note, deriving (4) from (15) requires an extra line or two. In my opinion it would greatly improve the clarity of the paper to use a more direct and straightforward derivation (perhaps with the interesting historical account of how the authors first derived this result given in an appendix). I could understand the more lengthy derivation being helpful if it gave insight into the source of variance reduction, but I don't see this personally, and the current discussion of variance reduction does not refer to the derivation of (15) at all.

The analysis of variance in section 2.6 leaves a lot to be desired. The central claim of the paper is that this method reduces variance, so it is an important section! Firstly, the variance of ARM vs AR is interesting, but the variance of ARM vs REINFORCE seems also highly relevant. Secondly, it seems like it would be very informative to look at the ratio of stdev to the mean for the ARM gradient estimate, since the true gradient is multiplied by sigmoid(phi) sigmoid(-phi) and so is very small if the probability of z = 1 is close to 0 or 1, exactly in the same regime where ARM has an advantage in variance reduction over AR. For example, it may be that learning in this regime is very difficult due to the weak gradient even if the estimate is extremely low variance. Thirdly and somewhat relatedly, in this same regime (z = 1 close to 0 or 1) the ARM gradient estimate is very often 0, meaning no learning takes place, so it seems a bit strange to argue that the new method is fantastic in the regime where it's almost always not learning! Of course, not learning is better than adding lots of spurious variance as reinforce would, but perhaps this could be made clearer. Finally, the theoretical analysis involving correlation gives very little insight and is extremely hand-wavy. A short worked example in the 1D or 2D case explicitly computing the variance of REINFORCE, AR and ARM seems like it would be highly informative.

Minor comments:

In the introduction, "*approximately* maximizing the marginal likelihood" might be more accurate, since as given in (28) the exact marginal likelihood is not optimized in practice, and the exact marginal likelihood is not of the form (1) but is rather the logarithm of something of the form (1).

I wasn't clear why "equal in distribution" was used a few things for things that are simply equal, such as just above (5).

In section 2.3, I don't see any real reason the estimates in (9) and (11) "could be highly positively correlated", other than an argument along the lines of the simple one given in section 2.6 that they're often equal and so zero.

As an aside, in section 3.1, it is great not to assume conditional independence of the binary latent variables across layers, but assuming conditional independence within each layer is still very restrictive. It is reasonable for the generative distribution to have this property, since the resulting net can still be essentially "universal" by stacking enough layers, but assuming this factorization in the variational distribution is highly restrictive with hard-to-reason-about consequences for the learned generative model. I realize this is a commonly used assumption and the authors are interested in the variance reduction properties of their approach rather than the training itself, but I just mention that it would be great to see extensions of the current work that can cope tractably with correlated latent variables within each layer.

In section 3.2, according to my understanding of standard terminology, "maximum likelihood inference" is a misnomer and would normally be "maximum likelihood estimation", since maximum likelihood is a method for estimating parameters whereas inference is about inferring latent variable values given parameters.

In section 4, it would be great to see some plots of explicit variance estimates of the different methods, given the overall goal of the paper (unless I just missed this?), even though figure 1 gives some insight into the variance characteristics.

In section 4.2, the expression log 1/K \sum_k Bernoulli... differs in the placement of log from Jang et al (2017). Which is the standard convention for this task?

---

> ### Author Response · Authors · 2018-11-13
> **Reverse-engineering the ARM estimator is leading to clearly simplified derivation and improved presentation**
>
>
> Thank you for your insightful and constructive comments and suggestions. Below please find a detailed point-by-point response.
>
> To Major Comment 1:
>
> We totally agree with you that it would be better to derive the univariate AR and ARM estimators from the analytic gradient via one dimensional integrations, and generalize them to multivariate ones using the law of total expectation. Motivated by your suggestion, we have reverse-engineered the proposed ARM estimator to considerably simplify its derivation. We have now derived the univariate AR (ARM) estimator with as few as two (three) equations, as shown in the revised Section 2.1, and have moved the original more complicated and longer derivation to the Appendix.
>
> To Major Comment 2:
>
> We have substantially expanded our analysis of variance in Section 2.3 of the revised paper.
>
> First, we have added in Proposition 2 the theoretical analysis of ARM vs REINFORCE for the univariate case, and a deeper theoretical analysis of ARM vs AR in Propositions 3 and 4 and Corollary 5. Empirically, we find the performance of AR is quite comparable to that of REINFORCE for all experiments, and we have now added the results of AR into Figures 1, 4, & 5 and Table 2 (a).
>
> Secondly, we have added the sample mean, sample stdev, and their ratio for the ARM and several other estimators in Figure 5, and added related discussions in the paragraph right before Section 4.1.
>
> Thirdly, we have provided  more discussion about the spiking behavior of the ARM gradient in the univariate case, in the first paragraph of page 7.
> We believe what ARM does in the univariate case is ``not learning at most of the iterations in order to occasionally move with big steps towards the right direction,'' which provides high gradient-to-noise ratio when it does move. By contrast, the REINFORCE and AR estimators are oscillating back and forth, as shown in Figure 1, resulting in ``learning all the time but frequently moving towards the wrong direction.''
>
> Fourthly, in addition to the newly added Propositions 2-4 and Corollary 5, we have added the plots of the empirical gradient variance into Figures 1, 3, and 5.

---

> > ### Author Response · Authors · 2018-11-25
> > **Response to AR3's minor comments**
> >
> >
> > To minor comments:
> >
> > Q1) In the introduction, "*approximately* maximizing the marginal likelihood" might be more accurate, since as given in (28) the exact marginal likelihood is not optimized in practice, and the exact marginal likelihood is not of the form (1) but is rather the logarithm of something of the form (1).
> >
> > A1: We agree with you and we have revised accordingly.
> >
> > Q2) In section 2.3, I don't see any real reason the estimates in (9) and (11) "could be highly positively correlated", other than an argument along the lines of the simple one given in section 2.6 that they're often equal and so zero.
> >
> > A2: The intuition is that if $f$ is always positive/negative, the scales of the two quantities $f(1_[\epsilon_1 e^{-\phi}< \epsilon_2])(1-\epsilon_1)$ and $f(1_[\epsilon_2 e^{-\phi}< \epsilon_1])(1-\epsilon_1)$ could be mainly influenced by the (1-\epsilon_1) term, which are shared by both quantities. To make it more concrete, in Proposition 3 and Appendix C of revised version, we have mathematically shown the variance of ARM with K Monte Carlo samples is lower than the AR estimator with 2K samples, suggesting the two quantities are positively correlated if $f$ is always positive/negative.
> >
> > Q3) As an aside, in section 3.1, it is great not to assume conditional independence of the binary latent variables across layers, but assuming conditional independence within each layer is still very restrictive. It is reasonable for the generative distribution to have this property, since the resulting net can still be essentially "universal" by stacking enough layers, but assuming this factorization in the variational distribution is highly restrictive with hard-to-reason-about consequences for the learned generative model. I realize this is a commonly used assumption and the authors are interested in the variance reduction properties of their approach rather than the training itself, but I just mention that it would be great to see extensions of the current work that can cope tractably with correlated latent variables within each layer.
> >
> > A3: Thank you for these nice suggestions! Currently we follow the common assumption that the latent variables in a certain layer given the latent variables from its upper layer are conditionally independent (marginally they are dependent). We will carefully consider possible extensions in our future research that can utilize correlated latent variables within each layer for the variational distribution.
> >
> > Q4) In section 4, it would be great to see some plots of explicit variance estimates of the different methods, given the overall goal of the paper (unless I just missed this?), even though figure 1 gives some insight into the variance characteristics.
> >
> > A4: We have added more gradient variance plots in Figures 1, 3, & 5.
> >
> > Q5) In section 4.2, the expression log 1/K \sum_k Bernoulli... differs in the placement of log from Jang et al (2017). Which is the standard convention for this task?
> >
> > A5: If sample h_k i.i.d from p(h \given x_u), by the law of large numbers, log ( 1/K \sum_k p(x_l \given h_k) ) will converge to the desired log marginal likelihood log( p(x_l \given x_u) ), which is a standard way to estimate log marginal likelihood, for example in “Importance weighted autoencoders (Burda et al., 2016)." Jang et al (2017) used 1/K \sum_k log(p(x_l \given h_k)) as the training target, but since Gumbel-softmax and ARM both use a single Monte Carlo sample based gradient estimate (i.e. K=1), the two expressions are the same as log(p(x_l \given h_1)) in practice.

---

> > > ### Comment · AnonReviewer3 · 2018-11-26
> > > **Thanks for all the improvements, which made the paper even clearer and stronger.**
> > >
> > > I really appreciate the authors taking my feedback on board in the the way they did. Section 2.1 in particular is now extremely clear and concise, and I think the paper is stronger because of it. The additional plots are also extremely helpful.
> > >
> > > A few minor comments:
> > >
> > > In a number of places the theoretical analysis presented assumes f is always positive (or negative). I would point out that this is quite artificial. In practice, when using say, REINFORCE, one would subtract at least a very approximate mean f value from f in order to reduce the gradient estimate variance.
> > >
> > > In section 2.1, "as originally derived" to me makes it sound like it was previously derived in another paper (especially if people aren't familiar with the references).
> > >
> > > Use of $\nu$ and $v$ in (8), (9), (10) seems unnecessarily confusing visually.
> > >
> > > In general it's extremely helpful for comparable plots for different systems to have the same axis scales. Is there any reason to use the differing scales in Figure 1? The difference in overall scale is often the main point, and easy to miss if all the plots are scaled differently. Also, why not use K = 100000 or something to get nice smooth plots of variance?
> > >
> > > Same comments about using a consistent scale for Figure 4 and Figure 5. And again, why not use K = very large for Figure 5 to get nice smooth plots (since it's essentially a theoretical analysis)? Also, plot SNR as the absolute value?

---

> > > > ### Author Response · Authors · 2018-11-27
> > > > **Thank you for your additional feedback and we have made revisions accordingly**
> > > >
> > > > We greatly appreciate your additional comments and suggestions. Below please find our point-by-point response:
> > > >
> > > > The positive (or negative) assumption on f is true if f is the ELBO or log-likelihood. While we sometimes make this assumption, mainly for the purpose of simplifying some of the theoretical analyses (e.g., guaranteeing there will always be variance reduction), we don't believe it is a critical requirement. We are extending the ARM estimator to several other applications, in some of which (such as reinforcement learning) f can take both positive and negative values, and we are further investigating how the property of f influences the performance. We hope we can get better insights and report our findings in future publications.
> > > >
> > > > We have changed "as originally derived by" to "as we initially derived it by" in the revised paper.
> > > >
> > > > We have changed $\nu$ to $j$ in (8)-(10).
> > > >
> > > > For the third row of Figure 1, we have now increased the samples to K=5000 to compute the empirical gradient variances. The reason that we do not use even more is because 1) the empirical variance plots now appear sufficiently smooth with K=5000, and 2) it already takes RELAX about 10 minutes to produce a single trace plot with K=5000, and use K=1000,000 would take a much longer time.
> > > >
> > > > We have now added the theoretical values, which can be viewed as the sample means with K = infinity, of the gradient variance into the updated Figure 1.
> > > >
> > > > We have modified the figures to use the same scale on the y-axis for better visual comparison, except if these values differ so significantly in scale that using the same scale may lose useful information.
> > > >
> > > > We have added the theoretical values, which can be viewed as the sample means with K = infinity, of the gradient standard deviation and SNR of REINFORCE, AR, and ARM into the updated Figure 5.
> > > >
> > > > We have now defined and plotted SNR as the absolute value in Figure 5.

---

### Author Response · Authors · 2018-11-25
**Summary of major improvements**

We thank the reviewers for their valuable comments and suggestions that have helped us to improve the paper. Listed below please find the major improvements we have made in our revision:

* Following the suggestions of AR3, we have now presented a much simplified derivation of the ARM estimator in the main body of the paper, and moved the original derivations to the appendix.

* Based on the feedback of AR2, we have now added the empirical gradient variance in Figure 1 for the toy example and in Figure 3 for the discrete latent variable model experiments.

* Following the comments of AR3 and AR2, we have more deeply investigated the variance reduction mechanism of the ARM estimator, and summarized our findings as Propositions 2-4 and Corollary 5, as shown in Section 2.3.

* Following the suggestions of AR1 and AR3, we have added more results in the toy example in Figures 1, 4, and 5, such as adding the results of the AR estimator and RELAX, and comparing signal-to-noise ratios between various gradient estimators.

Below please also find our point-by-point response to each reviewer's comments.

---

### Meta-Review · Area_Chair1 · 2018-12-11
**Good contribution, still a slog to read**

**Confidence:** 3
**Recommendation:** Accept (Poster)

**Metareview:**

This paper introduces a new way to estimate gradients of expectations of discrete random variables by introducing antithetic noise samples for use in a control variate.

Quality:  The experiments are mostly appropriate, although I disagree with the choice to present validation and test-set results instead of training-time results.  If the goal of the method is to reduce variance, then checking whether optimization is improved (training loss) is the most direct measure.  However reasonable people can disagree about this.

I also think the toy experiment (copied from the REBAR and RELAX paper) is a bit too easy for this method, since it relies on taking two antithetic samples.  I would have liked to see a categorical extension of the same experiment.

Clarity:  I think that this method will not have the impact it otherwise could because of the authors' fearless use of long equations and heavy notation throughout.  This is unavoidable to some degree, but
1) The title of the paper isn't very descriptive
2) Why not follow previous work and use \theta instead of \phi for the parameters being optimized?
The presentation has come a long way, but I fear that few besides our intrepid reviewers will have the stomach.  I recommend providing more intuition throughout.

Originality:  The use of antithetic samples to reduce variance is old, but this seems like a well-thought-through and non-trivial application of the idea to this setting.

Significance:  Ultimately I think this is a new direction in gradient estimators for discrete RVs.  I don't think this is the last word in this direction but it's both an empirical improvement, and will inspire further work.

---

> ### Author Response · Authors · 2019-02-01
> **Author response**
>
> We thank the AC for his/her comments. Below please find our response.
>
> We'd like to clarify that while we focus on presenting the test negative log-likelihoods/ELBOs in the tables, we have provided the trace plots of training and validation negative ELBOs in Figures 2, 6, 7.
>
> The toy experiment of maximizing E_z [(z-p_0)^2] indeed is very easy for the proposed ARM estimator, but becomes very challenging for REBAR and RELAX when p_0 approaches 0.5.
>
> We use $\phi$ for the parameters to avoid confusion for our experiments in discrete variational autoencoders, where $\phi$ is commonly used to denote the encoder parameter, while $\theta$ is commonly used to denote the decoder parameter. While the derivation may still appear complicated and heavy in notation, the actual implementation is in fact rather straightforward.
>
> Antithetic sampling for variance reduction is indeed old. However, antithetic sampling only becomes useful after performing variable augmentation and REINFORCE in the augmented space; without the Augment and REINFORCE steps, it is unclear how antithetic sampling can be applied to binary variables.
>
> Categorical extension of the binary ARM estimator involves much more sophisticated variable-swap and merge operations (much more notation heavy). We had a preliminary solution, which can be found in https://arxiv.org/abs/1807.11143 , and we have recently discovered another significantly improved solution. We plan to update that ArXiv submission in the near future.